# Machinability Assessment of Hybrid Nano Cutting Oil for Minimum Quantity Lubrication (MQL) in Hard Turning of 90CrSi Steel

**Tran Bao Ngoc [1], Tran Minh Duc [2], Ngo Minh Tuan [2], Vu Lai Hoang [3] and Tran The Long [2,*]**

[1]  Department of Fluids Mechanic, Faculty of Automotive and Power Machinery Engineering, Thai Nguyen University of Technology, Thai Nguyen 250000, Vietnam
[2]  Department of Manufacturing Engineering, Faculty of Mechanical Engineering, Thai Nguyen University of Technology, Thai Nguyen 250000, Vietnam
[3]  Department of Materials Engineering, Faculty of Mechanical Engineering, Thai Nguyen University of Technology, Thai Nguyen 250000, Vietnam
*  Correspondence: tranthelong@tnut.edu.vn or tranthelong90@gmail.com; Tel.: +84-985-288-777

**Abstract:** Friction and very high temperature are still the major challenges in hard machining technology and they greatly affect cutting efficiency. The application of the MQL (minimum quantity lubrication) method, using nanoparticles in order to improve the cooling lubrication performance of the base cutting oil, has proven to be a promising solution. Hence, this work aimed to investigate the effectiveness of $Al_2O_3/MoS_2$ hybrid nanofluid and $Al_2O_3$ and $MoS_2$ mono nanofluids in the hard turning of 90CrSi steel (60–62 HRC) under an MQL environment. The Box-Behnken experimental design was used for three input variables, including nanoparticle concentration, air pressure, and air flow rate. Their influences on surface roughness and cutting forces were studied. According to the obtained results, it was shown that the application of hybrid nano cutting oils in MQL contributes to achieving better hard machining performance than the use of mono nanofluids. In particular, a lower cutting temperature is reported and the values of surface roughness $R_a$, back force $F_p$, and cutting force $F_c$ were smaller and more stable under $Al_2O_3/MoS_2$ hybrid nanofluid MQL than those under $Al_2O_3$ and $MoS_2$ mono nanofluid MQL due to an improvement in cooling lubrication characteristics. Thus, this work provides a novel approach to study hybrid nanofluids for MQL hard machining.

**Keywords:** hard turning; surface roughness; cutting force; MQL; nano cutting oil; machinability assessment





## 1. Introduction

The problem of climate change has been increasingly present, greatly affecting people's lives around the world. This is posing new challenges to the manufacturing industry which must act faster and transform faster to respond to new, increasingly stricter environmental laws, and also to contribute to protecting the environment [1]. The metal cutting industry is also affected by this trend. It can be said that metal cutting processes are playing a very important role for the industry of each country, and this is one of the measures to evaluate the level of scientific and technological development. In the machining field, the use of cutting fluids in flood condition to lubricate and cool the contact zone is the most commonly used method; however, the treatment of these types of used lubricants has proven to be very expensive and their untreated discharge causes serious environmental pollution [2]. Therefore, the trends toward reducing dependence on industrial lubricants derived from mineral oil, switching to environmentally friendly cutting oils such as plant-based oils, or eliminating cutting oils, have been becoming more and more urgent, especially in the machining of hard materials. Machining in dry conditions, i.e., without using cutting oil, is a solution which has been researched and developed through the development of machine tools and cutting tool materials [3–5].

Today, higher-quality rigid machine tools combined with high-quality cutting tool materials have made it easier to directly cut high-hardness materials. However, the heat generated from the cutting zone in hard machining is very large, so it accelerates the tool wear rate, reduces the tool life, adversely affects the machined surface quality, and causes difficulties in handling the workpiece [4]. To overcome these problems, the machining of hard materials with minimal quantity lubrication (MQL) using nano cutting oils is a new solution, which significantly improves the lubrication and cooling efficiency in the cutting zone, thereby improving the cutting performance [6] as well as the cooling effect of the MQL technique [7]. In particular, this technology allows the use of vegetable oils with only a very small amount, so it has very environmentally friendly characteristics [1]. Li et al. [6] studied the heat transfer of $MoS_2$, $ZrO_2$, CNT, PCD, $Al_2O_3$, and $SiO_2$ palm-oil-based nanofluids when grinding Ni-based alloy. The authors found that the improvement in viscosity and thermal conductivity of the base vegetable oil was enhanced by suspending nanoparticles, thereby reducing the grinding forces and temperature. In addition, the higher nanoparticle concentration and smaller grain size would be more favorable for decreasing the cutting forces and improving the machined surface quality. Pryazhnikov et al. [8] pointed out that the thermal conductivity coefficient of nanofluids based on water, ethylene glycol, and engine oil was improved when compared to the base fluids. This factor is a complicated function influenced by the grain size, type, and concentration of nanoparticles as well as the types of base fluids. Ali et al. [9] carried out a study on the tribological characteristics of $Al_2O_3$ and $TiO_2$ nanoparticles suspended in automotive engine oil. The formation of laminating protective films created by $Al_2O_3$ nanoparticles and a decrease in the friction coefficient due to their rolling effect were discovered. Yıldırım et al. [10] studied the influence of hBN ester-based nanofluid on the MQL turning performance of Inconel 625. The authors found that better cutting efficiency, tool life, and surface quality were reported when compared to dry condition. Hegab et al. [11] studied MQL performance using multi-walled carbon nanotubes (MWCNTs) in ECOLUBRIC E200 vegetable oil in the turning process of Ti-6Al-4V alloy. The study results revealed that the cooling and lubricating effects of the MQL method, as well as the machined surface quality, were significantly improved, leading to a reduction in tool wear. They also carried out an investigation of tool performance and chip morphology in the turning of Inconel 718 using MWCNTs and $Al_2O_3$ nanofluid. The obtained findings showed that the deformed chip thickness was lower and that a more favorable chip morphology was formed because of the improvement in the MQL cooling and lubricating capabilities [12,13]. Uysal et al. [14] investigated the tool wear and surface roughness in milling under $MoS_2$ vegetable-based nanofluid MQL conditions. The excellent lubricating effect of $MoS_2$ nanosheets contributed to reducing the tool wear and improving the surface roughness. Eltaggaz et al. [15] carried out a study on the turning of austempered ductile iron (ADI) under an MQL environment using $Al_2O_3$ vegetable-oil-based nanofluid. A better cutting efficiency by using $Al_2O_3$ nanofluid MQL was reported when compared to dry, flood, and MQL with pure based oil. The main reasons for that were the improvement of thermal conductivity and the significant reduction of the friction coefficient.

These effects were achieved due to $Al_2O_3$ nanoparticles having high hardness, nearly spherical morphology, and good thermal conductivity [16]. When $Al_2O_3$ nanoparticles penetrate into the cutting zone, the "ball rolling" mechanism at the contact surfaces and the tribo film forming are the main factors in reducing the coefficient of friction [17]. In addition, $MoS_2$ nanoparticles have a sheet structure and possess a very small coefficient of friction, so they have a very good lubricating effect [18]. Therefore, this type of nanoparticle is also widely used in the MQL technique for metal cutting processes. Ayşegül Yücel and his co-authors [19] studied the influence of $MoS_2$ mineral-based nanofluid MQL on the turning performance of AA 2024 T3 aluminum alloy. The authors found that significant improvements in surface roughness and surface topography were achieved, and the built-up-edge (BUE) formation was eliminated by nanofluid MQL (NF MQL) conditions due having better lubricating effects than those of dry cutting. Furthermore, a lower tool wear rate and cutting

temperature were reported when machining Al alloy under NF MQL when compared to dry and pure MQL environments. The main reason for this phenomenon was the formation of tribo-film layer at the tool–chip interface. In recent years, there has been a trend toward using a combination of two types of nanoparticles suspended in the mineral cutting oil to take advantage of each nanoparticle type in order to improve cooling and lubricating efficiency in the cutting zone. Singh et al. [20] studied the effects of the MQL method using $Al_2O_3$-graphene hybrid nanofluid on the hard turning process. The cooling and lubricating efficiency of emulsion-based oil was enhanced, which contributed to improving the hard-turning performance, leading to a reduction in the values of surface roughness, cutting forces, and tool wear. Şirin et al. [21] investigated the effects of three different vegetable-based hybrid nanofluids, including hexagonal boron nitride (hBN)/graphite (Grpt), hBN/$MoS_2$, and Grpt/$MoS_2$, at different cutting speeds and feed rates. The obtained results indicated that hBN/Grpt hybrid nanofluid produced better results in terms of cutting forces, cutting temperature, surface roughness/topography, tool wear, and tool life. Zhang et al. [22] evaluated the lubricating effect of $MoS_2$/CNT hybrid nanofluid with synthetic lipids as the base fluid on Ni-based alloy grinding under an MQL environment. The authors integrated $MoS_2$ nanoparticles with good lubrication properties and CNTs with a high heat-conductivity coefficient. The results revealed that the $MoS_2$/CNT hybrid nanofluid achieved better lubrication performance than the nanofluid in terms of surface quality and grinding force. Dubey et al. [23] compared the effectiveness of $Al_2O_3$/graphene hybrid nanofluid to $Al_2O_3$ nanofluid with a biodegradable base oil in an MQL turning process of AISI 304 steel. The authors found that the average reduction in cutting forces, surface roughness, and cutting temperature with $Al_2O_3$/graphene hybrid nanofluid were 13%, 31%, and 14%, respectively, when compared to $Al_2O_3$ nanofluid. The nanoparticle concentration had a stronger impact on the cutting force than on the surface characteristics. Yıldırım et al. [24] compared the tribological behavior of hybrid with biodegradable-based nanofluids, including MWCNT/$Al_2O_3$, $Al_2O_3$/$MoS_2$, and MWCNT/$MoS_2$ in different concentration ratios to the mono nanofluids, for the hard turning of AISI 420 (55.5 HRC) using coated cermet tools. The obtained findings indicated that the hybrid nanofluids achieved better surface roughness than mono nanofluids, pure fluids, and dry cutting. The $Al_2O_3$/$MoS_2$ hybrid nanofluid produced the best surface topography. The cutting temperature and flank wear were significantly reduced by using hybrid nanofluid MQL when compared to pure MQL. Furthermore, the mono and hybrid nanofluids contributed to a significant prevention of failure modes for coated cermet cutting tools in hard turning as compared to dry and pure MQL. On the other hand, the use of naturally biodegradable, vegetable-based cutting oils is a new trend in the application of NF MQL technology, which is an important development toward sustainable production [21].

As seen in this literature analysis, the application of nanofluids as the cutting oil has become very popular in the metal cutting field. However, there has been very little information in these studies on the effects of using $Al_2O_3$ and $MoS_2$ nanofluids and $Al_2O_3$/$MoS_2$ hybrid nanofluids with soybean-based fluid on the hard turning of 90CrSi steel. Based on this observation, $Al_2O_3$, $MoS_2$, and $Al_2O_3$/$MoS_2$ hybrid nanofluids were used in the hard turning of 90CrSi steel (60–62 HRC) with different nanoparticle concentrations, air pressures, and air flow rates. The responses studied included surface roughness, surface microstructure, back force $F_p$, and cutting force $F_c$.

## 2. Material and Method

### 2.1. Experimental Set Up

Figure 1 shows the set up for hard turning experiments. CBN inserts with the designation of CCGW 09T308S 01020 FWH were utilized. The NOGA MiniCool MC1700 (Noga Engineering & Technology (2008) ltd, Shlomi, Israel) was used for the MQL system. The force-measure device was a 9257BA dynamometer (Kistler Instruments (Pte) Ltd., Midview, Singapore). The SJ-210 Mitutoyo surface roughness tester was used for measuring the surface roughness $R_a$ values with the cut-off length of 0.08 mm, measuring speed of

0.25 mm/s, and the retraction speed of the probe of 1 mm/s. After each cutting trial, the surface roughness was measured three times and taken as the average value. A KEYENCE VHX-7000 Digital Microscope (KEYENCE Corporation, Osaka, Japan) was used to investigate the surface microstructure and chip morphology. $Al_2O_3$ and $MoS_2$ nanoparticles with grain size of 30 nm were suspended in the soybean oil to form the nano/hybrid nano cutting oils in 1 h using an Ultrasons-HD ultrasonic homogenizer (JP Selecta, Abrera (Barcelona), Spain) at 40 kHz frequency and 600 W maximum power to ensure uniform distribution [16]. Hardened 90CrSi steel (60–62 HRC) with the diameter of 40 mm was utilized, and its chemical composition is shown in Table 1.

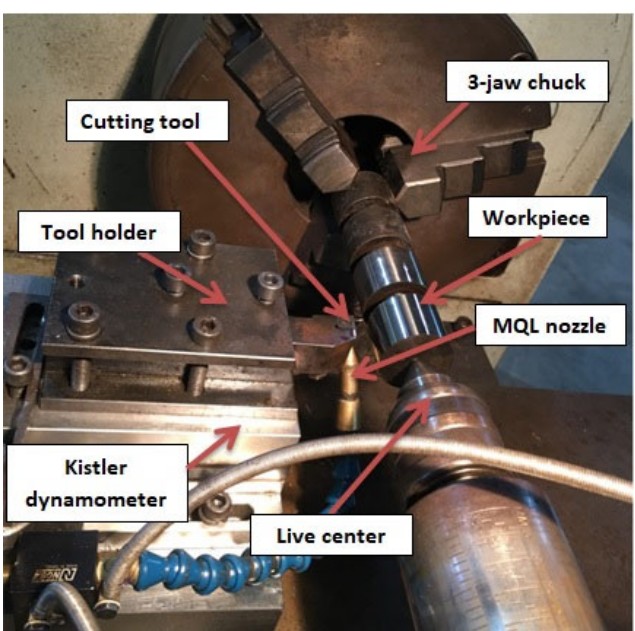

**Figure 1.** The experimental set up.

**Table 1.** Chemical composition in % of 90CrSi steel [25].

| Element | C | Si | Mn | Ni | S | P | Cr | Mo | W | V | Ti | Cu |
|---|---|---|---|---|---|---|---|---|---|---|---|---|
| **Weight (%)** | 0.85–0.95 | 1.20–1.60 | 0.30–0.60 | Max 0.40 | Max 0.03 | Max 0.03 | 0.95–1.25 | Max 0.20 | Max 0.20 | Max 0.15 | Max 0.03 | Max 0.3 |

*2.2. Experiment Design*

The $Al_2O_3$ and $MoS_2$ nanofluids and the $Al_2O_3$:$MoS_2$ (8:2) hybrid nanofluid were prepared with soybean oil (Cai Lan Vegetable Oil Co., Ltd., Ha Long City, Vietnam) as the base oil. Minitab 19.0 software was utilized for Box-Behnken experimental design with three input variables (nanoparticle concentration, air pressure, and air flow rate) and their levels, which were based on previous studies [25,26] (Table 2). A total number of 15 trials were conducted independently in triplicates. The cutting parameters were fixed at cutting speed $v_c$ = 160 m/min, feed rate f = 0.12 mm/rev., and cutting depth $a_p$ = 0.12 mm [24]. The responses included the surface roughness $R_a$, back force $F_p$, cutting force $F_c$, and chip color. Tables 3–5 summarize the experiment design with test run order and the measured values for surface roughness $R_a$, back force $F_p$, and cutting force $F_c$ under $MoS_2$ NF MQL, $Al_2O_3$ NF MQL, and $Al_2O_3$/$MoS_2$ hybrid nanofluid MQL (HF MQL), respectively. Each of the trials was repeated three times under the same cutting conditions and the average values were taken. The sampling frequency of cutting forces was 0.001 s. The cutting length for each experiment under the presented conditions was 83.73 m.

**Table 2.** Input variables and their levels.

| Input Variables | Unit | Symbol | Level | |
|---|---|---|---|---|
| | | | Low | High |
| Nanoparticle concentration of $Al_2O_3$ and $Al_2O_3/MoS_2$ | wt.% | NC | 0.5 | 1.5 |
| Nanoparticle concentration of $MoS_2$ | wt.% | NC | 0.2 | 0.8 |
| Air pressure | bar | p | 4 | 6 |
| Air flow rate | l/min | Q | 150 | 250 |

**Table 3.** The experiment design with test run order and the measured values under $MoS_2$ NF MQL.

| Std Order | Run Order | Input Variables | | | Responses | | |
|---|---|---|---|---|---|---|---|
| | | NC (wt.%) | p (bar) | Q (l/min) | $R_a$ (µm) | $F_p$ (N) | $F_c$ (N) |
| 1 | 8 | 0.2 | 4 | 200 | 0.249 | 220.6 | 98.7 |
| 2 | 5 | 0.8 | 4 | 200 | 0.355 | 508.7 | 112.9 |
| 3 | 11 | 0.2 | 6 | 200 | 0.356 | 345.5 | 108.9 |
| 4 | 12 | 0.8 | 6 | 200 | 0.385 | 510.9 | 111.5 |
| 5 | 13 | 0.2 | 5 | 150 | 0.345 | 268.0 | 97.1 |
| 6 | 6 | 0.8 | 5 | 150 | 0.442 | 460.2 | 108.4 |
| 7 | 10 | 0.2 | 5 | 250 | 0.240 | 236.0 | 97.3 |
| 8 | 4 | 0.8 | 5 | 250 | 0.318 | 468.2 | 124.3 |
| 9 | 2 | 0.5 | 4 | 150 | 0.254 | 101.4 | 83.7 |
| 10 | 7 | 0.5 | 6 | 150 | 0.280 | 135.9 | 85.1 |
| 11 | 14 | 0.5 | 4 | 250 | 0.276 | 165.6 | 92.3 |
| 12 | 9 | 0.5 | 6 | 250 | 0.274 | 143.2 | 94.4 |
| 13 | 15 | 0.5 | 5 | 200 | 0.244 | 124.8 | 105.3 |
| 14 | 3 | 0.5 | 5 | 200 | 0.238 | 104.8 | 84.6 |
| 15 | 1 | 0.5 | 5 | 200 | 0.250 | 104.4 | 85.5 |

**Table 4.** The experiment design with test run order and the measured values under $Al_2O_3$ NF MQL.

| Std Order | Run Order | Input Variables | | | Responses | | |
|---|---|---|---|---|---|---|---|
| | | NC (wt.%) | p (bar) | Q (l/min) | $R_a$ (µm) | $F_p$ (N) | $F_c$ (N) |
| 1 | 1 | 0.5 | 4 | 200 | 0.313 | 157.0 | 100.5 |
| 2 | 9 | 1.5 | 4 | 200 | 0.353 | 135.4 | 115.8 |
| 3 | 7 | 0.5 | 6 | 200 | 0.315 | 248.0 | 116.6 |
| 4 | 11 | 1.5 | 6 | 200 | 0.390 | 266.8 | 130.5 |
| 5 | 14 | 0.5 | 5 | 150 | 0.317 | 280.8 | 148.9 |
| 6 | 4 | 1.5 | 5 | 150 | 0.313 | 145.9 | 94.8 |
| 7 | 3 | 0.5 | 5 | 250 | 0.307 | 171.0 | 110.4 |
| 8 | 10 | 1.5 | 5 | 250 | 0.365 | 236.8 | 121.8 |
| 9 | 13 | 1 | 4 | 150 | 0.302 | 230.6 | 134.7 |
| 10 | 12 | 1 | 6 | 150 | 0.375 | 213.3 | 111.0 |
| 11 | 2 | 1 | 4 | 250 | 0.305 | 105.6 | 92.5 |
| 12 | 8 | 1 | 6 | 250 | 0.328 | 262.0 | 122.0 |
| 13 | 15 | 1 | 5 | 200 | 0.301 | 188.0 | 113.1 |
| 14 | 5 | 1 | 5 | 200 | 0.303 | 109.9 | 92.2 |
| 15 | 6 | 1 | 5 | 200 | 0.295 | 118.9 | 98.5 |

**Table 5.** The experiment design with test run order and the measured values under $Al_2O_3/MoS_2$ hybrid nanofluid MQL.

| Std Order | Run Order | Input Variables | | | Responses | | |
|---|---|---|---|---|---|---|---|
| | | *NC* (wt.%) | *p* (bar) | *Q* (l/min) | $R_a$ (µm) | $F_p$ (N) | $F_c$ (N) |
| 1 | 1 | 0.5 | 4 | 200 | 0.313 | 137.6 | 113.3 |
| 2 | 9 | 1.5 | 4 | 200 | 0.353 | 141.7 | 130.3 |
| 3 | 7 | 0.5 | 6 | 200 | 0.315 | 139.2 | 119.3 |
| 4 | 11 | 1.5 | 6 | 200 | 0.390 | 145.9 | 128.7 |
| 5 | 14 | 0.5 | 5 | 150 | 0.317 | 159.7 | 128.8 |
| 6 | 4 | 1.5 | 5 | 150 | 0.313 | 130.3 | 108.7 |
| 7 | 3 | 0.5 | 5 | 250 | 0.307 | 113.8 | 99.5 |
| 8 | 10 | 1.5 | 5 | 250 | 0.365 | 125.2 | 104.1 |
| 9 | 13 | 1 | 4 | 150 | 0.302 | 152.0 | 134.7 |
| 10 | 12 | 1 | 6 | 150 | 0.375 | 153.1 | 139.4 |
| 11 | 2 | 1 | 4 | 250 | 0.305 | 148.1 | 120.1 |
| 12 | 8 | 1 | 6 | 250 | 0.328 | 148.4 | 119.1 |
| 13 | 15 | 1 | 5 | 200 | 0.301 | 115.4 | 96.8 |
| 14 | 5 | 1 | 5 | 200 | 0.303 | 117.3 | 91.7 |
| 15 | 6 | 1 | 5 | 200 | 0.295 | 118.0 | 95.3 |

## 3. Results and Discussion

### 3.1. Surface Roughness $R_a$

Figure 2 depicts the effects of the cooling and lubricating methods, including $Al_2O_3/MoS_2$ HF MQL, $Al_2O_3$ NF MQL, and $MoS_2$ NF MQL, on surface roughness $R_a$, where the horizontal axis represents the order of experimental points in the standard order in experimental designs (Tables 3–5), and the vertical axis is the measured values of $R_a$. From Figure 2, it can be seen that using $Al_2O_3/MoS_2$ HF MQL (blue line) produced the smaller and the most stable surface roughness $R_a$ values. In the case of using $Al_2O_3$ NF MQL (red line), the surface roughness values and their amplitude of fluctuation were larger than those for the $Al_2O_3/MoS_2$ HF MQL conditions. The roughness values under $MoS_2$ NF MQL (black line) had the largest fluctuation amplitude at the experimental points from the first to eighth, while at the experimental points 9 to 15, the roughness values had small variation and were equivalent to those under the $Al_2O_3/MoS_2$ HF MQL environment. At each experimental point, the effect of using different cutting oils ($MoS_2$ NF MQL, $Al_2O_3$ NF MQL, and $Al_2O_3/MoS_2$ HF MQL) on the roughness value Ra was compared on the basis of the same regime of NC, p, and Q. Through the investigation of all 15 experimental points, the influence of the survey factors (NC, p, and Q) on the roughness value $R_a$ was evaluated.

The Pareto chart for the effects of standardized factors including NC, p, and Q on the surface roughness value $R_a$ is shown in Figure 3. The horizontal position of the limit line for the area where the inversion hypothesis is rejected was 2.571. The factors extending to the right beyond the limit line had strong influences on the responses.

Effect of nanoparticle concentration (NC): From the Pareto charts in Figure 3, it can be seen that nanoparticle concentration had the greatest effect on $R_a$ when using $Al_2O_3$ NF MQL (Figure 3b), followed by $MoS_2$ NF MQL (Figure 3a), and the least effect when using HF MQL (Figure 3c).

Effect of air pressure (p): For the three different nano cutting oils, air pressure had little effect on the roughness value $R_a$, on which the effect of using $Al_2O_3$ NF MQL was greatest (Figure 3b), followed by $MoS_2$ NF MQL (Figure 3a), and HF MQL was least (Figure 3c).

Effect of air flow rate (Q): The air flow rate Q had little effect on the roughness value $R_a$, on which the greatest effect was reported in the case of using $MoS_2$ NF MQL (Figure 3a), followed by HF MQL (Figure 3c), and then $Al_2O_3$ NF MQL (Figure 3b).

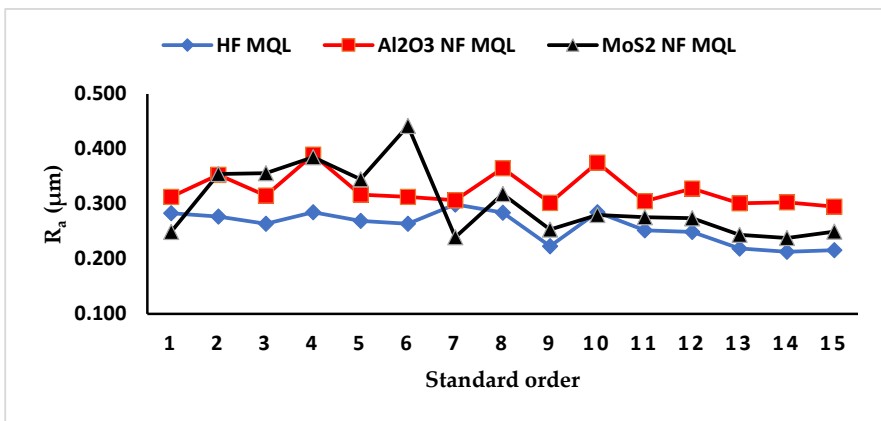

**Figure 2.** Effects of cooling and lubricating methods on surface roughness $R_a$ ($Al_2O_3/MoS_2$ hybrid nanofluid MQL: HF MQL; $Al_2O_3$ nanofluid MQL: $Al_2O_3$ NF MQL; $MoS_2$ nanofluid MQL: $MoS_2$ NF MQL).

| Std order | | 1 | 2 | 3 | 4 | 5 | 6 | 7 | 8 | 9 | 10 | 11 | 12 | 13 | 14 | 15 |
|---|---|---|---|---|---|---|---|---|---|---|---|---|---|---|---|---|
| NC | $Al_2O_3$ | 0.5 | 1.5 | 0.5 | 1.5 | 0.5 | 1.5 | 0.5 | 1.5 | 1.0 | 1.0 | 1.0 | 1.0 | 1.0 | 1.0 | 1.0 |
| (%) | $MoS_2$ | 0.2 | 0.8 | 0.2 | 0.8 | 0.2 | 0.8 | 0.2 | 0.8 | 0.5 | 0.5 | 0.5 | 0.5 | 0.5 | 0.5 | 0.5 |
| | $Al_2O_3/MoS_2$ | 0.5 | 1.5 | 0.5 | 1.5 | 0.5 | 1.5 | 0.5 | 1.5 | 1.0 | 1.0 | 1.0 | 1.0 | 1.0 | 1.0 | 1.0 |
| p (bar) | | 4 | 4 | 6 | 6 | 5 | 5 | 5 | 5 | 4 | 6 | 4 | 6 | 5 | 5 | 5 |
| Q (l/min) | | 200 | 200 | 200 | 200 | 150 | 150 | 250 | 250 | 150 | 150 | 250 | 250 | 200 | 200 | 200 |

The interaction effect between the input factors had little influence on $R_a$ except for the influence of the quadratic interaction of nanoparticle concentration (A*A) in Figure 3a,c. The results of analysis of variance are shown in Tables A1–A9 in the Appendix A. The suitability of the experimental model for the collected experimental data was evaluated through the coefficient of determination $R^2$. The $R^2$ coefficients for $MoS_2$ NF MQL, $Al_2O_3$ NF MQL, and $Al_2O_3/MoS_2$ HF MQL were 82.98%, 85.77%, and 89.78%, respectively, which proved that the experimental models were suitable for the experimental results.

For all three types of nano cutting oil, it was shown that at the experimental points from 9 to 15, the values and fluctuation amplitude of $R_a$ were smaller than at the experimental points from 1 to 8, and they were less dependent on the nanoparticle concentration and air pressure. This can be explained by the fact that at the experimental points from 9 to 15, there were medium levels of nanoparticle concentration, at 1.0% for the $Al_2O_3/MoS_2$ hybrid nano cutting oil and $Al_2O_3$ nano cutting oil and 0.5% for the $MoS_2$ nano cutting oil. This was the range of appropriate concentrations to produce a small and stable $R_a$ value [17,18]. For the other experimental points (1 to 8), either the low nanoparticle concentration was not enough to create the tribo film, or the high concentration caused the adhesive phenomena on the cutting edge to scratch the machined surface in the case of $MoS_2$ [27], or the mutual compression, collision, and impedance for $Al_2O_3$ nanoparticles, thereby producing inconsistent lubrication and negatively affecting the surface quality [26]. Particularly for $Al_2O_3/MoS_2$ HF MQL, because the concentrations of $Al_2O_3$ and $MoS_2$ nanoparticles were in the reasonable ranges, there was a combination between the two main lubrication mechanisms of the "roller" mechanism from $Al_2O_3$ nanoparticles and the "tribo film" formation of $MoS_2$ nanoparticles, so the surface roughness values were small and stable [28]. The obtained results showed that the reasonable concentration of nanoparticles in cutting oil will reduce the influences of air pressure *p* and air flow rate *Q*. This observation has great practical significance when it is necessary to quickly solve the practical problem, because it is only needed to choose the nanoparticle concentration in a reasonable range, and the air pressure and flow rate can be rapidly selected. However, in order to calculate specifically and accurately, it is still necessary to solve the optimization

problem. If the evaluation criterion is the surface roughness value $R_a$, the results showed that the $Al_2O_3/MoS_2$ HF MQL condition produced better results than $Al_2O_3$ NF MQL and $MoS_2$NF MQL because the $R_a$ values were smaller and more stable. However, for $MoS_2$ NF MQL, when the appropriate concentration of $MoS_2$ nanoparticles of about 0.5% was used, the results were equivalent to $Al_2O_3/MoS_2$ HF MQL.

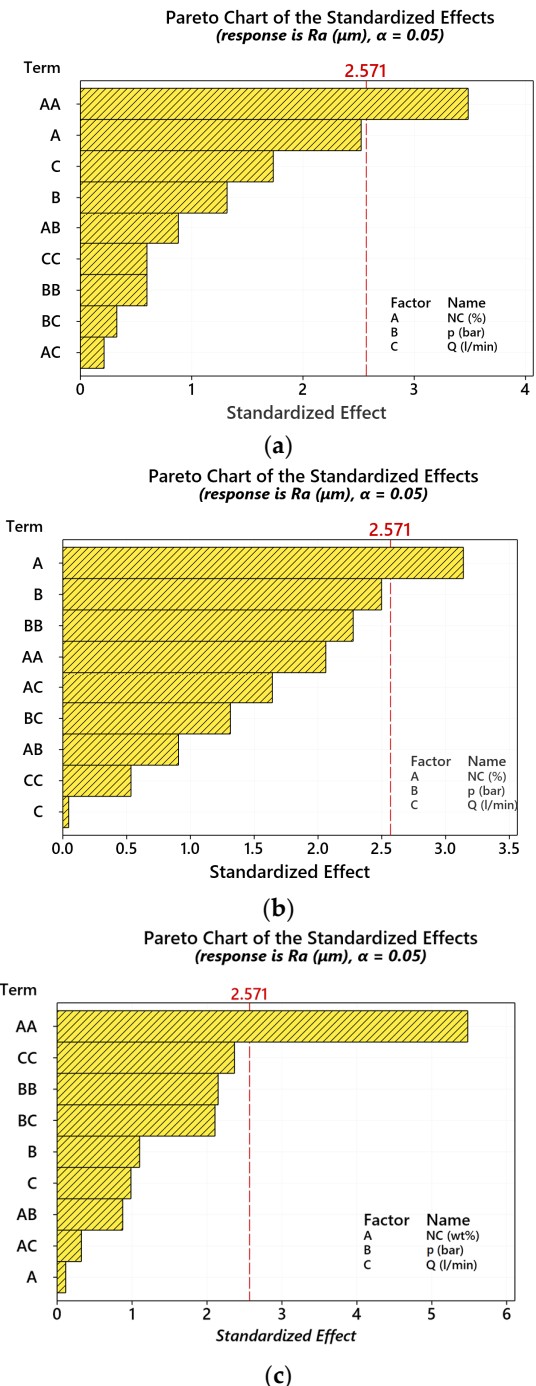

**Figure 3.** Pareto charts of the effects of standardized factors on surface roughness $R_a$ under: (**a**) $MoS_2$ NF MQL, (**b**) $Al_2O_3$ NF MQL, (**c**) $Al_2O_3/MoS_2$ HF MQL.

### 3.2. Surface Microstructure

To further evaluate the effectiveness of $Al_2O_3/MoS_2$ HF MQL, we used images of the machined surface microstructure and surface profile taken at three experimental points corresponding to 0.5% $MoS_2$ NF MQL, 1% $Al_2O_3$NF MQL, and 1% $Al_2O_3/MoS_2$ HF MQL,

with fixed cutting conditions and medium levels of air pressure, p = 5 bar, and air flow rate, Q = 200 l/min. Figure 4 shows the microstructure images and Figure 5 shows a 3D model and profile images of the machined surface in hard turning under different cooling lubrication conditions. Figure 4a is the microstructure of the machined surface with 0.5% $MoS_2$ NF MQL. It can be observed that the surface has a dark color with small purple stripes, and is smooth. The main reasons for these phenomena were the unique characteristics of $MoS_2$ nanoparticles created from their hexagonal crystal system combining Mo and S elements through short covalent bonds, but the space between them is large. Hence, the bond between two adjacent sulfur atom layers is weak, and the so-called "easy-to-slide planes" are generated by cutting forces to effectively reduce the friction coefficient on the contact faces, so it reduces the $R_a$ values [18,29,30]. Furthermore, the dark purple and light blue stains indicated the higher cutting temperature in the cutting zone [31]. On the other hand, similar to graphene nanosheets, $MoS_2$ nanosheets contribute to the formation of the tribo films owing to the good wettability, so the tribological effect is improved [32]. As a result, the surface profile is quite smooth, and the roughness values are small.

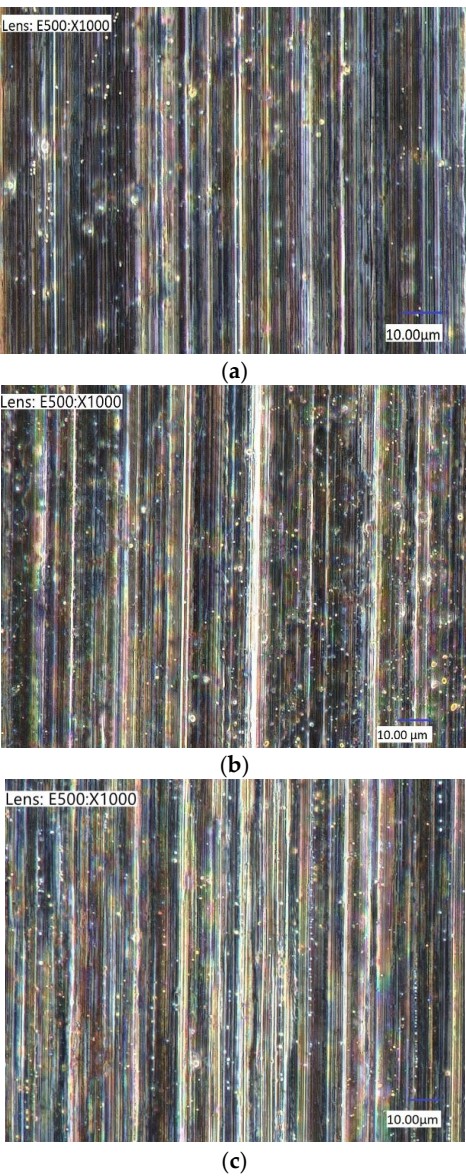

(a)

(b)

(c)

**Figure 4.** Microstructure and profile images of the machined surface under: (**a**) 0.5% $MoS_2$ NF MQL; (**b**) 1% $Al_2O_3$ NF MQL; (**c**) 1% $Al_2O_3$/$MoS_2$ HF MQL.

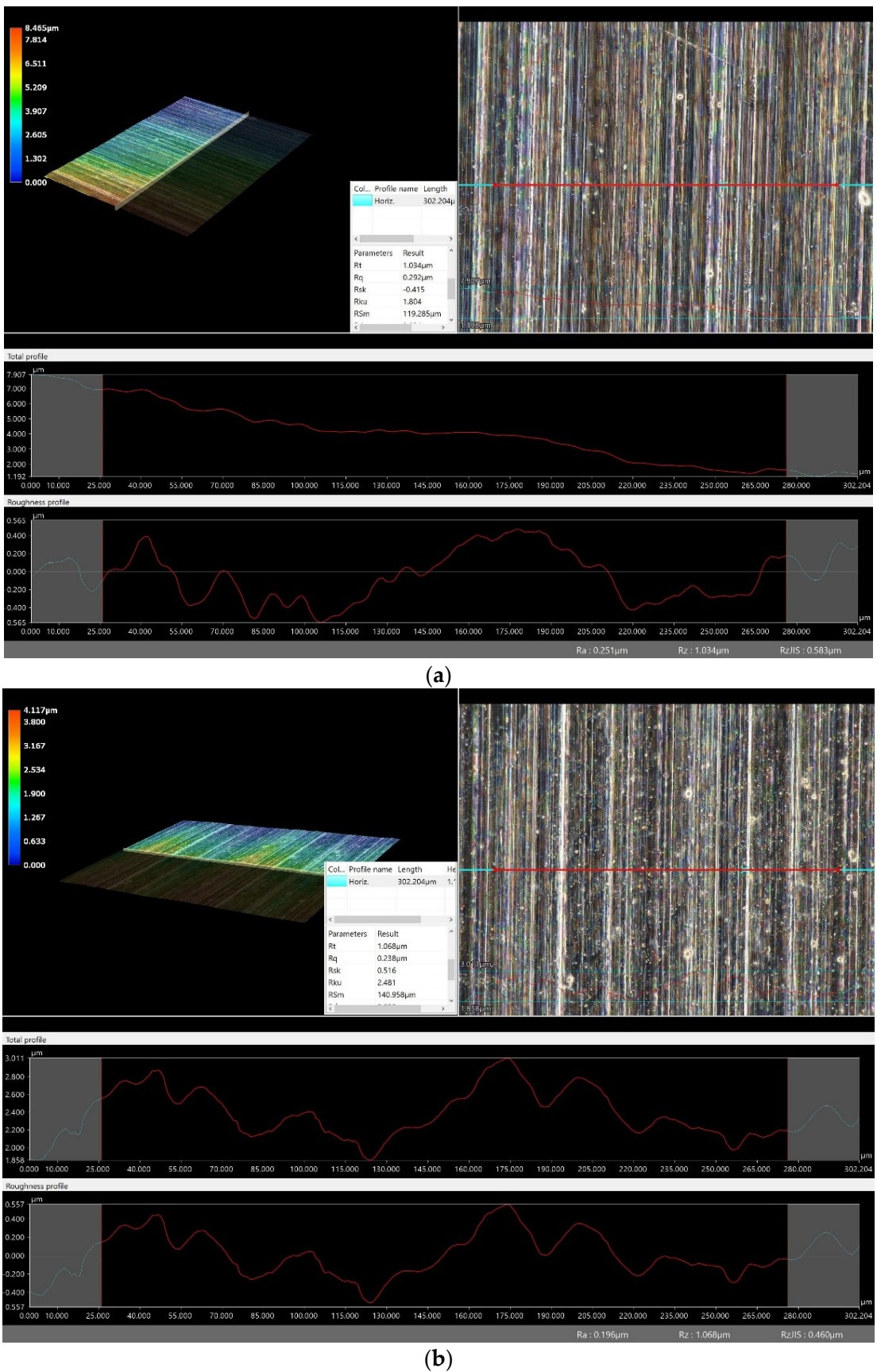

(**a**)

(**b**)

**Figure 5.** *Cont.*

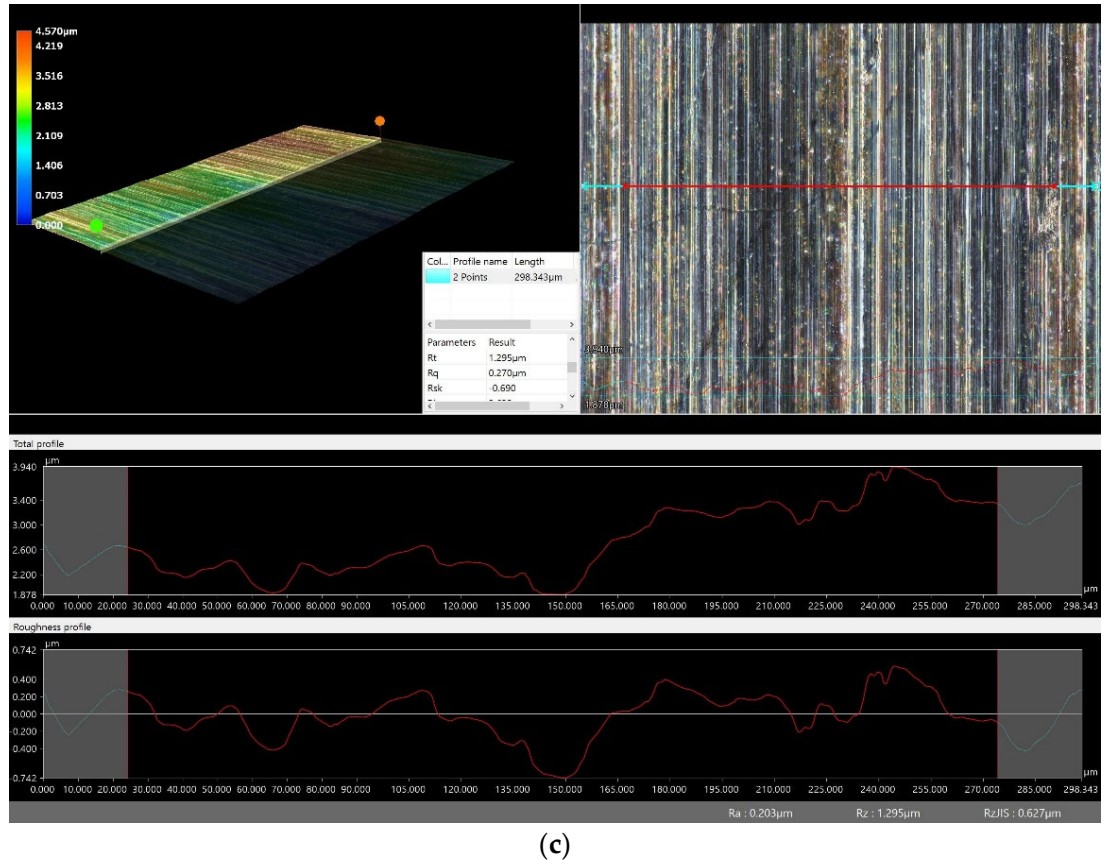

(**c**)

**Figure 5.** 3D and profile images of the machined surface under: (**a**) 0.5% $MoS_2$ NF MQL; (**b**) 1% $Al_2O_3$ NF MQL; (**c**) 1% $Al_2O_3/MoS_2$ HF MQL.

Figure 4b shows the surface microstructure under 1% $Al_2O_3$ NF MQL. The machined surface was quite smooth with white stripes and brown and purple-brown stains. The main reason for this was that $Al_2O_3$ nanoparticles have high hardness and nearly spherical morphology, so they create the "roller" effect in the contact zone to help reduce friction and cutting forces, contributing to a decrease in surface roughness [17]. In addition, due to the great pressure at the contact faces, the $Al_2O_3$ nanoparticles are pinched, deformed, and adhered to the machined surface, which is reflected in the white stripes in Figure 4b [26]. Figure 4c exhibits the surface microstructure when using 1% $Al_2O_3/MoS_2$ HF MQL. The machined surface was quite bright and smooth. The good friction-reducing effect of $MoS_2$ nanosheets combined with the roller and mending effects of $Al_2O_3$ nanoparticles contributed to a reduction in the cutting forces, surface roughness, and cutting temperature. Accordingly, the surface stains were mainly light straw color to dark straw color [28,32].

From the 3D-model and profile images of the machined surface in the different surveyed lubrication conditions, as shown in Figure 5, it can be seen that the machined surface was very smooth with small roughness values. In addition, there were no sharp differences when observing the surface profiles.

### 3.3. Effects on the Cutting Force Components $F_p$, $F_c$

The influences of the survey variables (nanoparticle concentration, air pressure *p*, air flow rate *Q*) on the cutting force $F_c$ are shown in Figure 6. The results showed that the three types of $Al_2O_3$, $MoS_2$, and $Al_2O_3/MoS_2$ nano cutting oil contributed to achieving small force values, and their changes were almost the same. There was no difference when changing the type of cutting oil. The effects of the investigated variables on the back force $F_p$ are shown in Figure 7. The values and variation of the back force $F_p$ depended heavily on the cooling lubrication modes and on the survey parameters. The use of the MQL method

with nano and hybrid nano cutting oils significantly reduces the friction between the flank face of the cutting tool and the machined face, so it mainly affects the back force $F_p$, and it has little effect on the feed force $F_f$ and cutting force $F_c$. Therefore, it is recommended to use as the criterion the back force $F_p$ rather than $F_c$ to evaluate the effect of the cooling and lubricating condition, which is quite convenient and obvious.

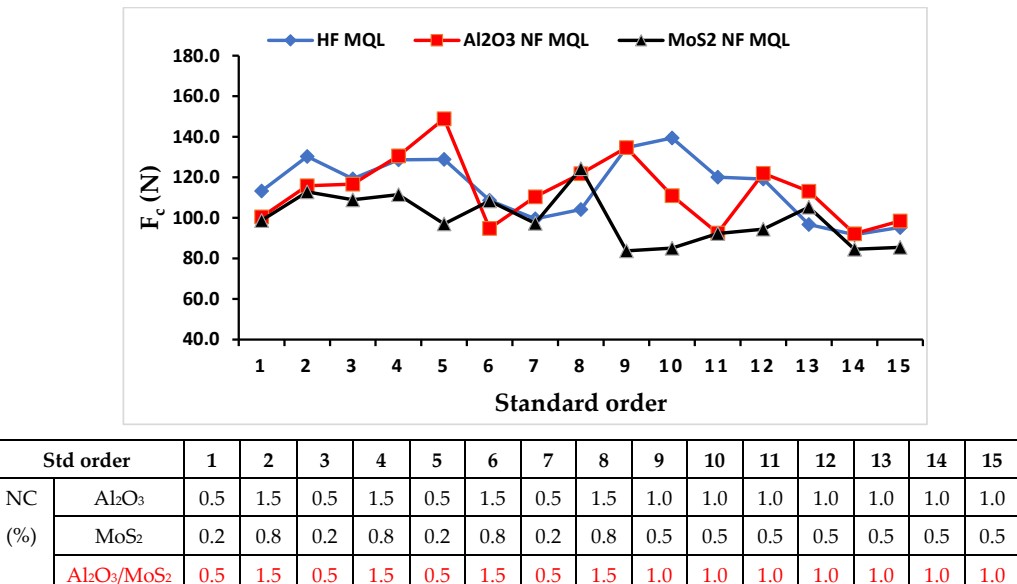

| Std order | | 1 | 2 | 3 | 4 | 5 | 6 | 7 | 8 | 9 | 10 | 11 | 12 | 13 | 14 | 15 |
|---|---|---|---|---|---|---|---|---|---|---|---|---|---|---|---|---|
| NC | Al₂O₃ | 0.5 | 1.5 | 0.5 | 1.5 | 0.5 | 1.5 | 0.5 | 1.5 | 1.0 | 1.0 | 1.0 | 1.0 | 1.0 | 1.0 | 1.0 |
| (%) | MoS₂ | 0.2 | 0.8 | 0.2 | 0.8 | 0.2 | 0.8 | 0.2 | 0.8 | 0.5 | 0.5 | 0.5 | 0.5 | 0.5 | 0.5 | 0.5 |
| | Al₂O₃/MoS₂ | 0.5 | 1.5 | 0.5 | 1.5 | 0.5 | 1.5 | 0.5 | 1.5 | 1.0 | 1.0 | 1.0 | 1.0 | 1.0 | 1.0 | 1.0 |
| p (bar) | | 4 | 4 | 6 | 6 | 5 | 5 | 5 | 5 | 4 | 6 | 4 | 6 | 5 | 5 | 5 |
| Q (l/min) | | 200 | 200 | 200 | 200 | 150 | 150 | 250 | 250 | 150 | 150 | 250 | 250 | 200 | 200 | 200 |

**Figure 6.** Effects of investigated variables on the cutting force $F_c$ (Al₂O₃/MoS₂ hybrid nanofluid MQL: HF MQL; Al₂O₃ nanofluid MQL: Al₂O₃ NF MQL; MoS₂ nanofluid MQL: MoS₂ NF MQL).

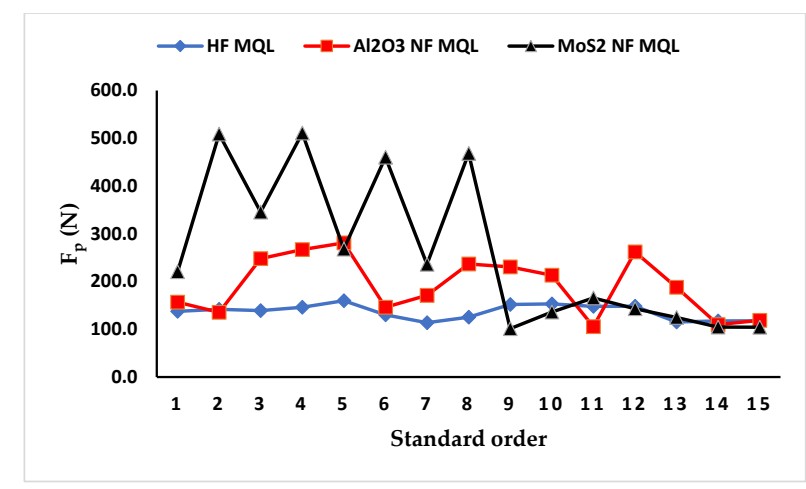

| Std order | | 1 | 2 | 3 | 4 | 5 | 6 | 7 | 8 | 9 | 10 | 11 | 12 | 13 | 14 | 15 |
|---|---|---|---|---|---|---|---|---|---|---|---|---|---|---|---|---|
| NC | Al₂O₃ | 0.5 | 1.5 | 0.5 | 1.5 | 0.5 | 1.5 | 0.5 | 1.5 | 1.0 | 1.0 | 1.0 | 1.0 | 1.0 | 1.0 | 1.0 |
| (%) | MoS₂ | 0.2 | 0.8 | 0.2 | 0.8 | 0.2 | 0.8 | 0.2 | 0.8 | 0.5 | 0.5 | 0.5 | 0.5 | 0.5 | 0.5 | 0.5 |
| | Al₂O₃/MoS₂ | 0.5 | 1.5 | 0.5 | 1.5 | 0.5 | 1.5 | 0.5 | 1.5 | 1.0 | 1.0 | 1.0 | 1.0 | 1.0 | 1.0 | 1.0 |
| p (bar) | | 4 | 4 | 6 | 6 | 5 | 5 | 5 | 5 | 4 | 6 | 4 | 6 | 5 | 5 | 5 |
| Q (l/min) | | 200 | 200 | 200 | 200 | 150 | 150 | 250 | 250 | 150 | 150 | 250 | 250 | 200 | 200 | 200 |

**Figure 7.** Effects of investigated variables on the back force $F_p$ (Al₂O₃/MoS₂ hybrid nanofluid MQL: HF MQL; Al₂O₃ nanofluid MQL: Al₂O₃ NF MQL; MoS₂ nanofluid MQL: MoS₂ NF MQL).

The MoS$_2$ NF MQL condition produced the F$_p$ values with the largest fluctuation amplitude (Figure 7). At the experimental points from 1 to 8, the back force F$_p$ was large and varied strongly. For the experimental points from 9 to 15, the values of the back force F$_p$ were small and stable because there was a reasonable concentration of MoS$_2$ nanoparticles (0.5%) to promote the cooling lubrication efficiency, so it produced stable cutting force values and good surface quality [18,27].

For the Al$_2$O$_3$ NF MQL mode, the F$_p$ values and their fluctuation amplitude were smaller than those of MoS$_2$ NF MQL but larger than those under Al$_2$O$_3$/MoS$_2$ HF MQL. The back force F$_p$ under the Al$_2$O$_3$/MoS$_2$ HF MQL environment was the smallest and most stable among the investigated cooling lubrication conditions. The reason was that the combination of the two types of nanoparticles to form the hybrid nanofluid promoted the advantages of both nanoparticle types. With an Al$_2$O$_3$/MoS$_2$ mixing ratio of 8:2 and a concentration of hybrid nanofluid from 0.5% to 1.5%, the concentration of each nanoparticle type was in the reasonable domain, so the back force F$_p$ was small and stable. Moreover, when the back force F$_p$ changed significantly, it caused vibration and adversely affected the machined part quality [33,34]. Hence, F$_p$ could be used as the criterion for evaluating the effectiveness of different cooling lubrication methods.

The Pareto charts of the effects on the cutting force F$_c$ (Figure 8) showed that all three investigated factors (*NC, p,* and *Q*) under the MoS$_2$ NF MQL and Al$_2$O$_3$ NF MQL environments had little effect on F$_c$. In Figure 8a, nanoparticle concentration (*NC*) had the greatest influence, followed by air flow rate (*Q*) and air pressure (*p*). For Al$_2$O$_3$ NF MQL in Figure 8b, the air flow rate (*Q*) had the strongest effect, followed by the air pressure (*p*) and finally the nanoparticle concentration (NC). For Al$_2$O$_3$/MoS$_2$ HF MQL (Figure 8c), the influence of the survey factors was greater, among which *Q* had the greatest influence, followed by *NC* and *p* with little influence. These observations were reflected in the results of the analysis of variance shown in Tables A4–A6 of the Appendix A. The R$^2$ coefficients for MoS$_2$ NF MQL, Al$_2$O$_3$ NF MQL, and Al$_2$O$_3$/MoS$_2$ HF MQL were 84.03%, 73.56%, and 92.93%, respectively, which proved that the experimental models were suitable for the experimental results. The use of the NF MQL and HF MQL methods mainly improved the frictional condition between the flank face of the cutting tool and the machined surface, so they had little effect on the cutting force component F$_c$. Hence, there was not much difference between the three different cooling lubrication methods, so the cutting force F$_c$ should not be used as the criterion for evaluation.

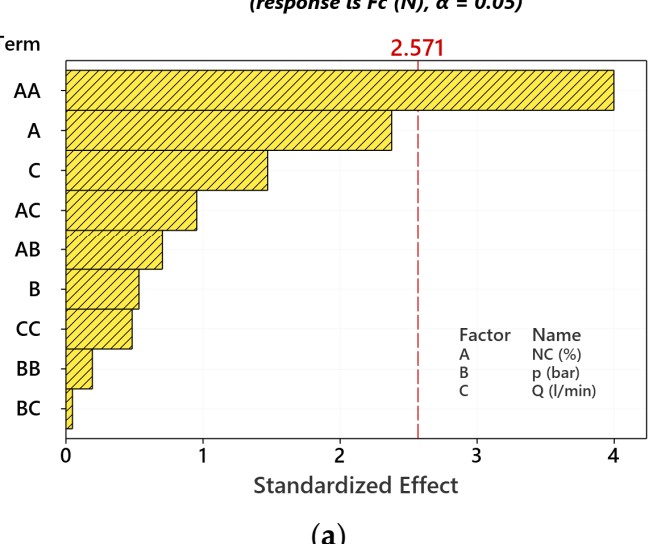

(**a**)

**Figure 8.** *Cont.*

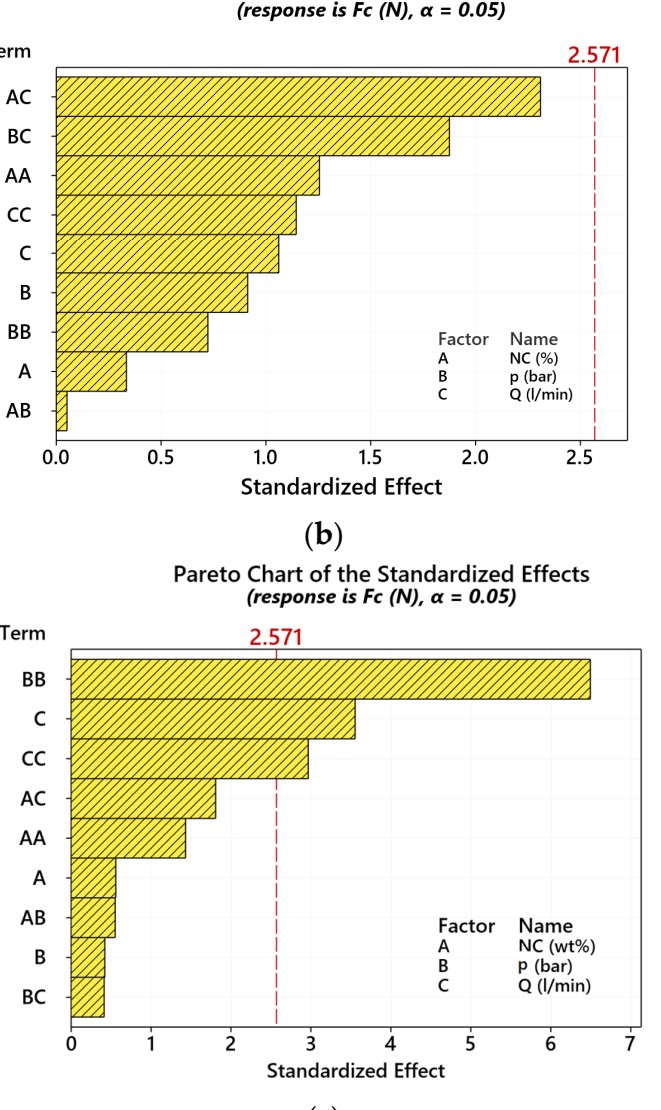

**Figure 8.** Pareto charts of the effects of standardized factors on cutting force $F_c$ under: (**a**) MoS$_2$ NF MQL, (**b**) Al$_2$O$_3$ NF MQL, (**c**) Al$_2$O$_3$/MoS$_2$ HF MQL.

The influence of cooling lubrication methods on the back force $F_p$ is shown in Figure 9. The influence of the survey factors on $F_p$ was evaluated through the Pareto charts. In the case of MoS$_2$ NF MQL (Figure 9a), the concentration of nanoparticles (*NC*) greatly affected $F_p$, followed by the air pressure (*p*) and air flow rate (*Q*). When using Al$_2$O$_3$ NF MQL (Figure 9b), *p* had the greatest influence, followed by *Q* and then *NC*. For HF MQL (Figure 9c), *Q* had the greatest influence, while *NC* and *p* had little effect. These effects were also evaluated through the results of the analysis of variance shown in Tables A7–A9 of the Appendix A. The $R^2$ coefficients for MoS$_2$ NF MQL, Al$_2$O$_3$ NF MQL, and Al$_2$O$_3$/MoS$_2$ HF MQL were 99.09%, 89.47%, and 89.85%, respectively, which proved that the experimental models were suitable for the experimental results.

With the three cooling lubrication conditions, the use of HF MQL produced the best effects in terms of the smallest values for $F_p$ and its fluctuation, followed by Al$_2$O$_3$ NF MQL and finally MoS$_2$ NF MQL. However, when using a reasonable concentration of MoS$_2$ nanoparticles in the cutting oil (about 0.5%), MoS$_2$ NF MQL had the same results as Al$_2$O$_3$/MoS$_2$ HF MQL, which was similar to the case with surface roughness.

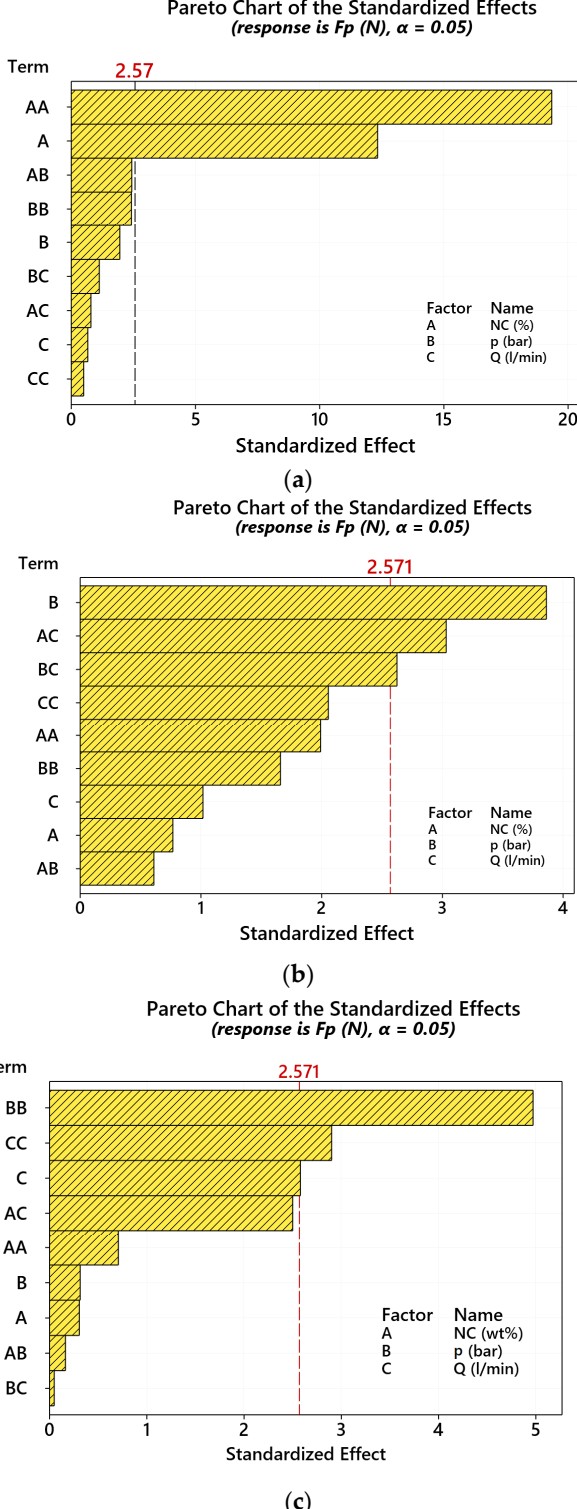

**Figure 9.** Pareto charts of the effects of standardized factors on cutting force $F_p$ under: (**a**) $MoS_2$ NF MQL, (**b**) $Al_2O_3$ NF MQL, (**c**) $Al_2O_3/MoS_2$ HF MQL.

## 4. Conclusions

The main target of this paper was to investigate the hard turning process of 90CrSi steel under an MQL environment using $Al_2O_3/MoS_2$ hybrid nanofluid, $Al_2O_3$ nanofluid, and $MoS_2$ nanofluid. Based on the Box-Behnken experimental design, the effects of nanoparticle concentration, air pressure, and air flow rate on surface roughness, back force $F_p$, and

cutting force $F_c$ were evaluated using Pareto charts. In addition, the surface microstructure was investigated.

The evaluation was carried out for all 15 experimental points according to the Box-Behnken experimental design. At each experimental point, comparisons and evaluations were carried out with the same cutting conditions: NC, p, and Q. With this new approach, the differences among the three different cooling lubrication methods were compared and evaluated, and the influence of the survey parameters on the objective functions was also considered.

The application of $Al_2O_3/MoS_2$ hybrid nano cutting oils in MQL improved the machining performance and produced better results when compared to MQL using the $MoS_2$ and $Al_2O_3$ mono nanofluids, as the values of the responses were the lowest and the most stable. For the investigated cooling lubrication conditions, the nanoparticle concentration and its quadratic interaction had the strongest effects on surface roughness $R_a$. The quadratic interactions of nanoparticle concentration (NC*NC) and air pressure (P*P) exhibited the largest impact on $F_p$, $F_c$ in the cases of $MoS_2$ nanofluid and $Al_2O_3/MoS_2$ hybrid nanofluid, while the interaction effect of nanoparticle concentration with air flow rate (NC*Q) and air pressure had a strong influence on $F_p$, $F_c$ when using $Al_2O_3$ nanofluid.

The $MoS_2$ nanoparticle concentration had strong effects on back force $F_p$. If a reasonable concentration value was selected (about 0.5%, the average studied range), it significantly reduced the influence of air pressure and air flow rate, and the values of the responses were small and stable. On the other hand, the back force $F_p$ could be used as the criterion for evaluating the effectiveness of different cooling lubrication methods.

In further work, more investigation is needed focusing on the optimization problems and to find more accurate values for nanoparticle concentration, air pressure, and air flow rate for NF MQL and HF MQL. Moreover, deeper study should be made of the machined surface microstructure to explore the cooling and lubricating mechanisms of nanoparticles in the cutting zone.

**Author Contributions:** Conceptualization, T.M.D. and T.T.L.; methodology, T.B.N., T.M.D., V.L.H. and T.T.L.; software, T.M.D., N.M.T. and V.L.H.; validation, T.M.D., N.M.T., V.L.H. and T.T.L.; formal analysis, T.B.N., N.M.T., V.L.H. and T.T.L.; investigation, T.B.N. and T.T.L.; resources, V.L.H.; data curation, N.M.T. and V.L.H.; writing—original draft, T.B.N., T.M.D., N.M.T. and T.T.L.; writing—review and editing, T.M.D. and T.T.L.; visualization, N.M.T. and V.L.H.; supervision, T.M.D. and T.T.L.; project administration, T.M.D. All authors have read and agreed to the published version of the manuscript.

**Funding:** This research was supported by Thai Nguyen University of Technology, Thai Nguyen University, Vietnam.

**Acknowledgments:** The work presented in this paper is supported by Thai Nguyen University of Technology, Thai Nguyen University, Vietnam.

**Conflicts of Interest:** The authors declare no conflict of interest.

## Appendix A

**Table A1.** Results of the ANOVA analysis of surface roughness $R_a$ under $MoS_2$ NF MQL.

| Source | DF | Adj SS | Adj MS | F-Value | *p*-Value |
|---|---|---|---|---|---|
| Model | 9 | 0.046012 | 0.005112 | 2.71 | 0.142 |
| Linear | 3 | 0.020964 | 0.006988 | 3.70 | 0.096 |
| NC | 1 | 0.012013 | 0.012013 | 6.36 | 0.053 |
| p | 1 | 0.003281 | 0.003281 | 1.74 | 0.245 |
| Q | 1 | 0.005671 | 0.005671 | 3.00 | 0.144 |
| Square | 3 | 0.023296 | 0.007765 | 4.11 | 0.081 |
| NC*NC | 1 | 0.022910 | 0.022910 | 12.14 | 0.018 |
| p*p | 1 | 0.000675 | 0.000675 | 0.36 | 0.576 |

**Table A1.** *Cont.*

| Source | DF | Adj SS | Adj MS | F-Value | *p*-Value |
|---|---|---|---|---|---|
| Q*Q | 1 | 0.000675 | 0.000675 | 0.36 | 0.576 |
| 2-Way Interaction | 3 | 0.001752 | 0.000584 | 0.31 | 0.818 |
| NC*p | 1 | 0.001463 | 0.001463 | 0.78 | 0.419 |
| NC*Q | 1 | 0.000086 | 0.000086 | 0.05 | 0.840 |
| p*Q | 1 | 0.000203 | 0.000203 | 0.11 | 0.756 |
| Error | 5 | 0.009437 | 0.001887 | | |
| Lack-of-Fit | 3 | 0.009371 | 0.003124 | 94.42 | 0.010 |
| Pure Error | 2 | 0.000066 | 0.000033 | | |
| Total | 14 | 0.055449 | | | |

"*" represents the interactions between the factors.

**Table A2.** Results of the ANOVA analysis of surface roughness $R_a$ under $Al_2O_3$ NF MQL.

| Source | DF | Adj SS | Adj MS | F-Value | *p*-Value |
|---|---|---|---|---|---|
| Model | 9 | 0.010922 | 0.001214 | 3.35 | 0.098 |
| Linear | 3 | 0.005832 | 0.001944 | 5.37 | 0.051 |
| NC | 1 | 0.003570 | 0.003570 | 9.85 | 0.026 |
| p | 1 | 0.002261 | 0.002261 | 6.24 | 0.055 |
| Q | 1 | 0.000001 | 0.000001 | 0.00 | 0.965 |
| Square | 3 | 0.003191 | 0.001064 | 2.94 | 0.138 |
| NC*NC | 1 | 0.001539 | 0.001539 | 4.25 | 0.094 |
| p*p | 1 | 0.001876 | 0.001876 | 5.18 | 0.072 |
| Q*Q | 1 | 0.000103 | 0.000103 | 0.29 | 0.616 |
| 2-Way Interaction | 3 | 0.001899 | 0.000633 | 1.75 | 0.273 |
| NC*p | 1 | 0.000298 | 0.000298 | 0.82 | 0.406 |
| NC*Q | 1 | 0.000977 | 0.000977 | 2.70 | 0.162 |
| p*Q | 1 | 0.000625 | 0.000625 | 1.72 | 0.246 |
| Error | 5 | 0.001812 | 0.000362 | | |
| Lack-of-Fit | 3 | 0.001777 | 0.000592 | 34.17 | 0.029 |
| Pure Error | 2 | 0.000035 | 0.000017 | | |
| Total | 14 | 0.012734 | | | |

"*" represents the interactions between the factors.

**Table A3.** Results of the ANOVA analysis of surface roughness $R_a$ under $Al_2O_3/MoS_2$ HF MQL.

| Source | DF | Adj SS | Adj MS | F-Value | *p*-Value |
|---|---|---|---|---|---|
| Model | 9 | 0.010447 | 0.001161 | 4.88 | 0.048 |
| Linear | 3 | 0.000522 | 0.000174 | 0.73 | 0.576 |
| NC | 1 | 0.000003 | 0.000003 | 0.01 | 0.913 |
| p | 1 | 0.000288 | 0.000288 | 1.21 | 0.321 |
| Q | 1 | 0.000231 | 0.000231 | 0.97 | 0.370 |
| Square | 3 | 0.008661 | 0.002887 | 12.13 | 0.010 |
| NC*NC | 1 | 0.007148 | 0.007148 | 30.04 | 0.003 |
| p*p | 1 | 0.001099 | 0.001099 | 4.62 | 0.084 |
| Q*Q | 1 | 0.001333 | 0.001333 | 5.60 | 0.064 |
| 2-Way Interaction | 3 | 0.001263 | 0.000421 | 1.77 | 0.269 |
| NC*p | 1 | 0.000182 | 0.000182 | 0.77 | 0.422 |
| NC*Q | 1 | 0.000025 | 0.000025 | 0.11 | 0.759 |
| p*Q | 1 | 0.001056 | 0.001056 | 4.44 | 0.089 |
| Error | 5 | 0.001190 | 0.000238 | | |
| Lack-of-Fit | 3 | 0.001172 | 0.000391 | 43.40 | 0.023 |
| Pure Error | 2 | 0.000018 | 0.000009 | | |
| Total | 14 | 0.011636 | | | |

"*" represents the interactions between the factors.

**Table A4.** Results of the ANOVA analysis of cutting force $F_c$ under $MoS_2$ NF MQL.

| Source | DF | Adj SS | Adj MS | F-Value | *p*-Value |
|---|---|---|---|---|---|
| Model | 9 | 1766.40 | 196.27 | 2.92 | 0.125 |
| Linear | 3 | 543.68 | 181.23 | 2.70 | 0.156 |
| NC | 1 | 379.09 | 379.09 | 5.65 | 0.063 |
| p | 1 | 19.07 | 19.07 | 0.28 | 0.617 |
| Q | 1 | 145.52 | 145.52 | 2.17 | 0.201 |
| Square | 3 | 1128.19 | 376.06 | 5.60 | 0.047 |
| NC*NC | 1 | 1072.68 | 1072.68 | 15.98 | 0.010 |
| p*p | 1 | 2.49 | 2.49 | 0.04 | 0.855 |
| Q*Q | 1 | 15.64 | 15.64 | 0.23 | 0.650 |
| 2-Way Interaction | 3 | 94.54 | 31.51 | 0.47 | 0.717 |
| NC*p | 1 | 33.24 | 33.24 | 0.50 | 0.513 |
| NC*Q | 1 | 61.15 | 61.15 | 0.91 | 0.384 |
| p*Q | 1 | 0.15 | 0.15 | 0.00 | 0.964 |
| Error | 5 | 335.69 | 67.14 | | |
| Lack-of-Fit | 3 | 62.02 | 20.67 | 0.15 | 0.921 |
| Pure Error | 2 | 273.67 | 136.84 | | |
| Total | 14 | 2102.09 | | | |

"*" represents the interactions between the factors.

**Table A5.** Results of the ANOVA analysis of cutting force $F_c$ under $Al_2O_3$ NF MQL.

| Source | DF | Adj SS | Adj MS | F-Value | *p*-Value |
|---|---|---|---|---|---|
| Model | 9 | 2796.93 | 310.77 | 1.55 | 0.329 |
| Linear | 3 | 416.50 | 138.83 | 0.69 | 0.596 |
| NC | 1 | 22.55 | 22.55 | 0.11 | 0.751 |
| p | 1 | 167.54 | 167.54 | 0.83 | 0.403 |
| Q | 1 | 226.42 | 226.42 | 1.13 | 0.337 |
| Square | 3 | 600.85 | 200.28 | 1.00 | 0.466 |
| NC*NC | 1 | 316.81 | 316.81 | 1.58 | 0.265 |
| p*p | 1 | 105.21 | 105.21 | 0.52 | 0.502 |
| Q*Q | 1 | 263.35 | 263.35 | 1.31 | 0.304 |
| 2-Way Interaction | 3 | 1779.58 | 593.19 | 2.95 | 0.137 |
| NC*p | 1 | 0.53 | 0.53 | 0.00 | 0.961 |
| NC*Q | 1 | 1072.56 | 1072.56 | 5.34 | 0.069 |
| p*Q | 1 | 706.50 | 706.50 | 3.51 | 0.120 |
| Error | 5 | 1005.15 | 201.03 | | |
| Lack-of-Fit | 3 | 773.97 | 257.99 | 2.23 | 0.324 |
| Pure Error | 2 | 231.18 | 115.59 | | |
| Total | 14 | 3802.08 | | | |

"*" represents the interactions between the factors.

**Table A6.** Results of the ANOVA analysis of cutting force $F_c$ under $Al_2O_3/MoS_2$ HF MQL.

| Source | DF | Adj SS | Adj MS | F-Value | *p*-Value |
|---|---|---|---|---|---|
| Model | 9 | 3085.15 | 342.79 | 7.31 | 0.021 |
| Linear | 3 | 615.22 | 205.07 | 4.37 | 0.073 |
| NC | 1 | 14.66 | 14.66 | 0.31 | 0.600 |
| p | 1 | 8.36 | 8.36 | 0.18 | 0.690 |
| Q | 1 | 592.20 | 592.20 | 12.62 | 0.016 |
| Square | 3 | 2294.11 | 764.70 | 16.30 | 0.005 |
| NC*NC | 1 | 96.30 | 96.30 | 2.05 | 0.211 |
| p*p | 1 | 1979.57 | 1979.57 | 42.19 | 0.001 |
| Q*Q | 1 | 413.08 | 413.08 | 8.80 | 0.031 |
| 2-Way Interaction | 3 | 175.82 | 58.61 | 1.25 | 0.385 |
| NC*p | 1 | 14.36 | 14.36 | 0.31 | 0.604 |
| NC*Q | 1 | 153.39 | 153.39 | 3.27 | 0.130 |
| p*Q | 1 | 8.07 | 8.07 | 0.17 | 0.696 |
| Error | 5 | 234.62 | 46.92 | | |
| Lack-of-Fit | 3 | 220.80 | 73.60 | 10.65 | 0.087 |
| Pure Error | 2 | 13.83 | 6.91 | | |
| Total | 14 | 3319.78 | | | |

"*" represents the interactions between the factors.

**Table A7.** Results of the ANOVA analysis of back force $F_p$ under $MoS_2$ NF MQL.

| Source | DF | Adj SS | Adj MS | F-Value | *p*-Value |
|---|---|---|---|---|---|
| Model | 9 | 344,220 | 38,247 | 60.37 | 0.000 |
| Linear | 3 | 99,040 | 33,013 | 52.11 | 0.000 |
| NC | 1 | 96,332 | 96,332 | 152.05 | 0.000 |
| p | 1 | 2427 | 2427 | 3.83 | 0.108 |
| Q | 1 | 282 | 282 | 0.44 | 0.535 |
| Square | 3 | 240,212 | 80,071 | 126.38 | 0.000 |
| NC*NC | 1 | 236,944 | 236,944 | 373.99 | 0.000 |
| p*p | 1 | 3724 | 3724 | 5.88 | 0.060 |
| Q*Q | 1 | 160 | 160 | 0.25 | 0.637 |
| 2-Way Interaction | 3 | 4968 | 1656 | 2.61 | 0.163 |
| NC*p | 1 | 3758 | 3758 | 5.93 | 0.059 |
| NC*Q | 1 | 400 | 400 | 0.63 | 0.463 |
| p*Q | 1 | 810 | 810 | 1.28 | 0.310 |
| Error | 5 | 3168 | 634 | | |
| Lack-of-Fit | 3 | 2896 | 965 | 7.11 | 0.126 |
| Pure Error | 2 | 271 | 136 | | |
| Total | 14 | 347,388 | | | |

"*" represents the interactions between the factors.

**Table A8.** Results of the ANOVA analysis of back force $F_p$ under $Al_2O_3$ NF MQL.

| Source | DF | Adj SS | Adj MS | F-Value | *p*-Value |
|---|---|---|---|---|---|
| Model | 9 | 46,527.0 | 5169.7 | 4.72 | 0.051 |
| Linear | 3 | 18,106.6 | 6035.5 | 5.51 | 0.048 |
| NC | 1 | 645.8 | 645.8 | 0.59 | 0.477 |
| p | 1 | 16,327.1 | 16,327.1 | 14.90 | 0.012 |
| Q | 1 | 1133.6 | 1133.6 | 1.03 | 0.356 |
| Square | 3 | 10,407.9 | 3469.3 | 3.17 | 0.123 |
| NC*NC | 1 | 4342.9 | 4342.9 | 3.96 | 0.103 |
| p*p | 1 | 3013.5 | 3013.5 | 2.75 | 0.158 |
| Q*Q | 1 | 4627.9 | 4627.9 | 4.22 | 0.095 |
| 2-Way Interaction | 3 | 18,012.5 | 6004.2 | 5.48 | 0.049 |
| NC*p | 1 | 408.2 | 408.2 | 0.37 | 0.568 |
| NC*Q | 1 | 10,063.1 | 10,063.1 | 9.19 | 0.029 |
| p*Q | 1 | 7541.2 | 7541.2 | 6.88 | 0.047 |
| Error | 5 | 5477.8 | 1095.6 | | |
| Lack-of-Fit | 3 | 1829.1 | 609.7 | 0.33 | 0.807 |
| Pure Error | 2 | 3648.7 | 1824.3 | | |
| Total | 14 | 52,004.8 | | | |

"*" represents the interactions between the factors.

**Table A9.** Results of the ANOVA analysis of back force $F_p$ under $Al_2O_3/MoS_2$ HF MQL.

| Source | DF | Adj SS | Adj MS | F-Value | *p*-Value |
|---|---|---|---|---|---|
| Model | 9 | 2959.31 | 328.81 | 4.92 | 0.047 |
| Linear | 3 | 458.10 | 152.70 | 2.28 | 0.197 |
| NC | 1 | 6.25 | 6.25 | 0.09 | 0.772 |
| p | 1 | 6.64 | 6.64 | 0.10 | 0.765 |
| Q | 1 | 445.21 | 445.21 | 6.66 | 0.049 |
| Square | 3 | 2081.26 | 693.75 | 10.37 | 0.014 |
| NC*NC | 1 | 33.47 | 33.47 | 0.50 | 0.511 |
| p*p | 1 | 1653.34 | 1653.34 | 24.72 | 0.004 |
| Q*Q | 1 | 562.55 | 562.55 | 8.41 | 0.034 |
| 2-Way Interaction | 3 | 419.95 | 139.98 | 2.09 | 0.220 |
| NC*p | 1 | 1.80 | 1.80 | 0.03 | 0.876 |
| NC*Q | 1 | 418.00 | 418.00 | 6.25 | 0.054 |
| p*Q | 1 | 0.16 | 0.16 | 0.00 | 0.963 |
| Error | 5 | 334.36 | 66.87 | | |
| Lack-of-Fit | 3 | 330.62 | 110.21 | 58.86 | 0.017 |
| Pure Error | 2 | 3.74 | 1.87 | | |
| Total | 14 | 3293.67 | | | |

"*" represents the interactions between the factors.

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
