# Peer review of "Machinability Assessment of Hybrid Nano Cutting Oil for Minimum Quantity Lubrication (MQL) in Hard Turning of 90CrSi Steel"

_lubricants, doi:10.3390/lubricants11020054_

Round 1

Reviewer 1 Report

The paper presents some interesting work on Machinability Assessment of Nano and Hybrid Nano Cutting Oil in MQL Hard Turning of 90CrSi Steel. The work is further supported by supported by analysis and some experimental results. However, the paper needs to undertake the following revisions:

(1) The paper is better titled as 'Machinability Assessment of Hybrid Nano Cutting Oil for Minimum Quantity Lubrication (MQL) in Hard Turning of 90CrSi Steel', it should avoid using abbreviation in the title.

(2) The keywords should better be rearranged and kept in six keywords as normally expected, such as 'Hard turning; surface roughness; cutting force; MQL; nano cutting oil; machinability assessment'.

(3) There should be a clear 'scale' mark (not just amplification) on the photos in Figures 3 and 4.

(4) In Section 3, the paper should provide the clarification and discussion on effects of nano cutting oil on the tool wear and material removal rate (MRR) in the hard turning. 

(5) Furthermore, the paper should provide some in-depth analysis and discussion on the tribological enhancement mechanism of using nano cutting in hard turning.

(6) The following very relevant paper in the topic area should be included in References section, particularly against the above comments (4) and (5):

Effect of self-developed graphene lubricant on tribological behaviour of silicon carbide/silicon nitride interface, Ceramics International 45 (8), 2019, 10211-10222 

Author Response

RESPONSES TO THE REVIEWER 1

We are very grateful for the reviews provided by the editors and each of the external reviewers of this manuscript. Please see below, our detailed response to comments.

The paper presents some interesting work on Machinability Assessment of Nano and Hybrid Nano Cutting Oil in MQL Hard Turning of 90CrSi Steel. The work is further supported by supported by analysis and some experimental results. However, the paper needs to undertake the following revisions:

(1) The paper is better titled as 'Machinability Assessment of Hybrid Nano Cutting Oil for Minimum Quantity Lubrication (MQL) in Hard Turning of 90CrSi Steel', it should avoid using abbreviation in the title.

Answer:

Thank you very much. The paper title was changed by following the sugguestion of the reviewer.

(2) The keywords should better be rearranged and kept in six keywords as normally expected, such as 'Hard turning; surface roughness; cutting force; MQL; nano cutting oil; machinability assessment'.

Answer:

Thank you very much. The keywords were revised and modified by following the sugguestion of the reviewer.

(3) There should be a clear 'scale' mark (not just amplification) on the photos in Figures 3 and 4.

Answer:

Thank you very much. The 'scale' marks were added on the photos in Figures 3 and 4.

(4) In Section 3, the paper should provide the clarification and discussion on effects of nano cutting oil on the tool wear and material removal rate (MRR) in the hard turning. 

Answer:

Thank you very much. Material removal rate (MRR) is not the aim of this article. We have investigated the tool wear and had deep analysis, but the authors would like to publish this content in the next work.

(5) Furthermore, the paper should provide some in-depth analysis and discussion on the tribological enhancement mechanism of using nano cutting in hard turning.

Answer:

Thank you very much. The deep analysis and discussion on the tribological enhancement mechanism of using nano cutting were added in the revised manuscript and marked in red color.

(6) The following very relevant paper in the topic area should be included in References section, particularly against the above comments (4) and (5):

- Effect of self-developed graphene lubricant on tribological behaviour of silicon carbide/silicon nitride interface, Ceramics International 45 (8), 2019, 10211-10222 

Answer:

Thank you very much for your valuable comment. The suggested paper was very useful to improve the deep analysis and was cited in Ref. 31 in the revised manuscript.

Reviewer 2 Report

Dear Author, I reviewed your manuscript. This work are focused on very important science field but the manuscript was prepared very weakly, especially in methodology chapter. 

1) Abstract does not report on the purpose of the study and its findings in a precise manner. 

2) The introduction should include a broader overview of the current state of knowledge on the application of MQL and nono and hybrid machining fluids

eg. 

Vineet, D., Sharma, K. A., Vats, P., Pimenov, D. Y., Giasin, K., & Chuchala, D. (2021). Study of a Multicriterion Decision-Making Approach to the MQL Turning of AISI 304 Steel Using Hybrid Nanocutting Fluid. Materials, 14, 7207. https://doi.org/10.3390/ma14237207

3) The methodology is very poorly prepared in fact the main elements of the experimental tests are not described. What is Ra, Fy, Fz, Kf ? If one measures cutting forces, one measures the specific component of the cutting force, not the direction of the dynamometer measurement. The scientist should know which component of the cutting force in his measurement correlates with which component of the measured force.

What is the Kf and why it is analysed. What does it contribute to the analysis of the problem ?

The symbols for the cutting parameters are missing. 

A description of how the surface roughness parameter is measured and relevant information on the measurement methodology are missing. 

The dimensions of the workpiece are not given. There is also a lack of sampling frequency of cutting forces and information on its selection in relation to the cutting parameters used. How was the signal filtered?  What was the measurement system's own frequency? 

4) The results are presented in a rough manner without standard deviations and statistical analyses. 

5) Conclusions are not structured in a precise way relating to the specific test results obtained

Author Response

RESPONSES TO THE REVIEWER 2

We are very grateful for the reviews provided by the editors and each of the external reviewers of this manuscript. Please see below, our detailed response to comments.

Dear Author, I reviewed your manuscript. This work are focused on very important science field but the manuscript was prepared very weakly, especially in methodology chapter. 

1) Abstract does not report on the purpose of the study and its findings in a precise manner. 

 Answer:

Thank you very much. The abstract was written by following the reviewer’s comment. The changes were in red color.

2) The introduction should include a broader overview of the current state of knowledge on the application of MQL and nono and hybrid machining fluids

  1.  

Vineet, D., Sharma, K. A., Vats, P., Pimenov, D. Y., Giasin, K., & Chuchala, D. (2021). Study of a Multicriterion Decision-Making Approach to the MQL Turning of AISI 304 Steel Using Hybrid Nanocutting Fluid. Materials, 14, 7207. https://doi.org/10.3390/ma14237207

 Answer:

Thank you very much. The work was revised carefully and cited in the revised manuscript to improve the introduction section.

3) The methodology is very poorly prepared in fact the main elements of the experimental tests are not described. What is Ra, Fy, Fz, Kf ? If one measures cutting forces, one measures the specific component of the cutting force, not the direction of the dynamometer measurement. The scientist should know which component of the cutting force in his measurement correlates with which component of the measured force.

What is the Kf and why it is analysed. What does it contribute to the analysis of the problem ?

Answer:

Thank you very much. Ra values with the cut-off length of 0.08 mm was measured by SJ-210 Mitutoyo surface roughness tester. After each cutting trial, the surface roughness was measured by three times and taken by the average values.

Thank you very much. Fy is the thrust cutting force component, and Fz is the tangential cutting force component. The cutting force coefficient Kf was calculated by .

Thank you very much for your valuable comment. The cutting force coefficient Kf was investigated in this study because it is one of the important parameters reflecting the machinability of the cutting tool and the technological error in machining. In particular, when the thrust force component Fy is large, the cutting force coefficient Kf will increase, which makes the actual depth of cut smaller than the required depth of cut. Hence, machining errors and vibrations will occur during the cutting process, adversely affecting the quality of the machined parts. This factor plays an even more important role, especially in the finishing process of heat treat-ed steels with high hardness, where high dimensional accuracy and surface quality are very demanding. The discussion was revised and improved.

4) The symbols for the cutting parameters are missing. 

Answer:

The symbols for the cutting parameters were added in the revised manuscript and marked in red color.

5) A description of how the surface roughness parameter is measured and relevant information on the measurement methodology are missing. 

Answer:

Thank you very much. Ra values with the cut-off length of 0.08 mm was measured by SJ-210 Mitutoyo surface roughness tester. After each cutting trial, the surface roughness was measured by three times and taken by the average values.

6) The dimensions of the workpiece are not given. There is also a lack of sampling frequency of cutting forces and information on its selection in relation to the cutting parameters used. How was the signal filtered?  What was the measurement system's own frequency? 

Answer:

Thank you very much. The dimensions of the workpiece was given in section 2.1. The sampling frequency of cutting forces was added. The cutting forces were directly measured during the cutting process by using 9257BA dynamometer connected to data acquisition system A/D DQA N16210 (made by National instruments, USA), and DASYlab 10.0 software installed on the computer. The reviewer can see in the following figures.

7) The results are presented in a rough manner without standard deviations and statistical analyses. 

Answer:

Thank you very much. The results and discussion were expanded to improve the paper quality. The changes were marked in red color. The standard deviations and statistical analyses were discussed in the previous work in Ref. 25, which is not the main scope in this paper. We can completely add to these analyses if necessary.

8) Conclusions are not structured in a precise way relating to the specific test results obtained

Answer:

Thank you very much. Conclusions were restructured and rewritten by following the reviewer’s comment.

Reviewer 3 Report

My comments are in the attached file

Author Response

RESPONSES TO THE REVIEWER 3

We are very grateful for the reviews provided by the editors and each of the external reviewers of this manuscript. Please see below, our detailed response to comments.

I consider that this work should be rejected, corrected and improved before being published in Lubricants.

My comments are as follows:

1) It is necessary to correct the English throughout the text.

Answer:

Thank you very much. The English language was revised throughout the paper and marked in red color.

2) When citing any work, indicate the last name of the first author (without initials). Then write "et al." if the number of authors is more than three (if there are two authors, both last names are written without initials) and then the citation. For example, Li et al. [6].

Answer:

Thank you very much. The citation was modified by following the reviewer’s comments.

3) the introduction should be rewritten, indicating the type of liquid lubricant used in each paper, since it is not clear if the cited papers also used soybean oil.

Answer:

Thank you very much. The introduction was rewritten and expanded. The type of liquid lubricant used in each paper was added in the revised manuscript in red color.

4) the last paragraph of the introduction does not explain that soybean oil has been implemented un the presentó work.

Answer:

Thank you very much. The use of soybean oil and the explaination were added in the revised manuscript in red color.

5) in the materials section it is not explained how the nano powders were mixed homogeneously in the cutting oil, explain.

Answer:

Thank you very much. Al2O3 and MoS2 nanoparticles with grain size of 30 nm were suspended in the soybean oil to form the nano/hybrid nano cutting oils in 1 hours using a Ultrasons-HD ultrasonic homogenizer (JP Selecta, Abrera (Barcelona), Spain) at 40 kHz frequency and 600 W maximum power to ensure the uniform distribution. The information was added in the revised manuscript.

6) there is no explanation of the composition of the cutting oil used: pure soybean oil, oil with water, their proportions, etc.

Answer:

Thank you very much. The cutting oil used in this paper is pure soybean oil.

7) the properties and manufacturer of soybean oil are not indicated.

Answer:

Thank you very much. The pure soybean oil is produced from natural materials with modern refining technology of Cai Lan Vegetable Oil Co., Ltd. in Vietnam, which is added in the revised manuscript.

8) It is not indicated how long the cutting process lasted for each experiment under the conditions presented (minutes, meters).

Answer:

The cutting length for each experiment under the presented conditions is 83.73 meters, which is added in the revised manuscript.

9) give an explanation of why the Box-Behnken was used for the design of the experiment, and how many repetitions of each experiments carried out had.

Answer:

Box-Benhen experimental design has can investigate 3 or more variables with smaller experiments than  the central composite design. With three factors the BB design by default will have three center points and is given in the Minitab output. The number of repetitions is 3, which is added in the revised manuscript.

10) line 158- .... Three input variables (list them)

Answer:

Thank you very much. Three input variables was listed in the revised manuscript.

11) considered that running 15 experiments for three different fluids and not giving a detailed and convincing analysis of the results obtained with the help of statistics is a great waste of time. There is no analysis of the influence of each of the parameters on each of the variables studied.

Answer:

Thank you very much. The purpose is to use the Box-Behnken optimal experimental design to evaluate the influence of the input variables on the resposes. This is the initial processing to compare hybrid and mono nanofluids, and use excel for comparison. The solution of this problem lays the foundation for solving the optimization problem. When solving the optimization problem, there will be a statistical analysis, which will be published in the next work.

12) Figures 2, 5, 6,7 are not informative, since they do not show any relationship between roughness, cutting forces and force coefficients with any controllable factor of the experiment.

Answer:

Thank you very much for your valuable comments. The comparison of all points in the experiment in Figures 2,5,6,7 aims to evaluate the efficiency and stability of lubrication when comparing nano-hybrid cutting oil with nano-oil under different nanoparticle concentration, air pressure, and air flow rate, especially on cutting force component Fy and cutting force coefficient Kf. These are two very important parameters that affect the technological dimensional error in hard machining, which distinguishs to cutting process of unheat-treated steels.

13) nowhere in the manuscript is it explained how the temperature of the chips was determined based on the color on their surface. Give an explanation, and some links about the method used by other authors, not yours.

Answer:

Thank you very much. Because it is difficult to measure the heat directly from the cutting process, especially in the cutting zone. Therefore, in order to evaluate the lubricating efficiency of nano-hybrid cutting oil compared to mono nano cutting oil, the authors used an indirect evaluation method through the color of the generated chips. The author uses the corresponding steel color and temperature table cited in Ref. 30.

 14) line 234 - how can a SEM image indicate what type of color each surface has?

Answer:

Thank you very much. The SEM images taken on KEYENCE VHX-7000 Digital Microscope can reflect the true color of the machined surface. The authors also changed the images with higher quality so that the color of the machined surface can be easily observed. The deeper discussion was provided and expanded in red color.

15) there is no explanation of why in figure 3 the surfaces obtained at 5 bar and a flow rate of 200 l/min were chosen? Why did you omit the surfaces obtained under other conditions?

Answer:

Thank you very much. Here the authors choose the average concentration of nanoparticles, air pressure and air flow rate because these are reasonable levels of these parameters. The authors added citations to the revised manuscript.

Reviewer 4 Report

The paper deals with Al2O3/MoS2 hybrid nanofluid and Al2O3/MoS2 mono nanofluid used to examine the hard turning performance of 90CrSi steel in a MQL environment. Surface roughness, cutting forces, and cutting force efficiency were examined. The authors have studied in previous work the use of nanoparticles for the same application so the novelty compared to the past is limited. The work is still a continuation of previous work. Some revision are necessary.

-Which parameters did you use to measure roughness (cut off, evaluation length) ? How many measurements were made for each condition?

-line 214: The authors show SEM images (figures 3 and 4) of the surfaces without writing in the experimental that it was used. Please add the specifications and put in the images the  scale marker and not the magnification. Furthermore, why did you choose to show these two conditions? explain why.

-In the part about the microscope images the authors talk about the different colors but only give an explanation of the white parts. Please explain the other colors as well.

Line 254.Please delete one “of” in the sentence.

Author Response

RESPONSES TO THE REVIEWER 4

We are very grateful for the reviews provided by the editors and each of the external reviewers of this manuscript. Please see below, our detailed response to comments.

The paper deals with Al2O3/MoS2 hybrid nanofluid and Al2O3/MoS2 mono nanofluid used to examine the hard turning performance of 90CrSi steel in a MQL environment. Surface roughness, cutting forces, and cutting force efficiency were examined. The authors have studied in previous work the use of nanoparticles for the same application so the novelty compared to the past is limited. The work is still a continuation of previous work. Some revision are necessary.

Answer:

Thank you very much for your very valuable comments. Compared to the previous work, although conducted on the same experimental set-up, the research content of the two articles is very different. In the previous article, we only aimed to investigate the overall effects of the input parameters including fluid type, nanoparticle type, cutting speed and depth of cut. The research results show the general influence trends and the interaction influence of the input factors on the cutting forces and surface roughness. In this article, we specifically investigate the influence of parameters such as nanoparticle concentration, air pressure and air flow rate when using mono nanofluid and hybrid nanofluid cutting oils. Evaluation parameters include surface roughness, cutting forces and cutting force coefficient, chip morphology and machined surface microstructure. Besides, in this study, CBN insert was used instead of carbide tools as in previous study.

  1. Which parameters did you use to measure roughness (cut off, evaluation length) ? How many measurements were made for each condition?

Answer:

Thank you very much. The surface roughness Ra was measured to investigate in this study. The cut-off length was added in the revised manuscript. After each cutting trial, the surface roughness was measured by three times and taken by the average values.

  1. line 214: The authors show SEM images (figures 3 and 4) of the surfaces without writing in the experimental that it was used. Please add the specifications and put in the images the scale marker and not the magnification. Furthermore, why did you choose to show these two conditions? explain why.

Answer:

Thank you very much for your very valuable comments. The scale markers for SEM images on Figures 3,4 were added in the revised manuscript.

  1. In the part about the microscope images the authors talk about the different colors but only give an explanation of the white parts. Please explain the other colors as well.

Answer:

Thank you very much for your very valuable comments. The quality of the microscope images was improved and the deeper explanation for other colors was added in the revised manuscript in red color.

  1. Line 254.Please delete one “of” in the sentence.

Answer:

Thank you very much. The sentence was revised and modified by following the suggestion of the reviewer.

Round 2

Reviewer 2 Report

Dear Authors, 

thank you for your answers and implemented corrections. However,  my main remarks were not corrected in the manuscript. 

What components of the resultant cutting force correspond to the measured force components? For the cutting process, we distinguish between three components of the resultant cutting force: cutting force Fc, feed force Ff, back force Fp. All components are very important for the analysis of the cutting process. I my opinion the proposed coefficient is not important and can by used only for discussion. 

Symbol of cutting speed is vc, not Vc. 

I got information from the authors that the roughness measurements were repeated 3 times for each sample. This information should be included in the manuscript. In addition, there were no repetitions of the cutting tests, which is not scientific and conclusions cannot be based on such results. 15 trials were performed, but each pass has different process parameters. Statistically, each cut should be repeated a minimum of 7 times for the results to be statistically reliable. 

Standard deviations must be included in the results presented and statistical analysis. That a similar analysis is in another article is, of course, mentionable. However, the previous article described a different study, so the results were also different. If this is not the case, is this self-plagiarism?

Author Response

RESPONSES TO THE REVIEWER 2

We are very grateful for the reviews provided by the editors and each of the external reviewers of this manuscript. Please see below, our detailed response to comments.

thank you for your answers and implemented corrections. However,  my main remarks were not corrected in the manuscript. 

1) What components of the resultant cutting force correspond to the measured force components? For the cutting process, we distinguish between three components of the resultant cutting force: cutting force Fc, feed force Ff, back force Fp. All components are very important for the analysis of the cutting process. I my opinion the proposed coefficient is not important and can by used only for discussion. 

Answer:

Thank you very much. The cutting force Fc and the back force Fp were clearifed and revised by following the reviewer’s suggestion. The proposed cutting coefficient was removed in the revised manuscript.

2) Symbol of cutting speed is vc, not Vc. 

Answer:

Thank you very much. The symbol of cutting speed was modified.

3) I got information from the authors that the roughness measurements were repeated 3 times for each sample. This information should be included in the manuscript. In addition, there were no repetitions of the cutting tests, which is not scientific and conclusions cannot be based on such results. 15 trials were performed, but each pass has different process parameters. Statistically, each cut should be repeated a minimum of 7 times for the results to be statistically reliable. 

Answer:

The roughness measurements were repeated 3 times for each sample. This information was added in the manuscript  and mard in red color.

Thank you very much for your valuable comment. The design of the experiment the author used here was based on the Box benken experimental design. Box-Behnken design is still considered to be more proficient and most powerful than other designs such as the three-level full factorial design, central composite design (CCD) and Doehlert design, despite its poor coverage of the corner of nonlinear design space. Also the central point in Box benken experimental design was repeated three times as shown in the experimental design in the manuscript.

Furthermore, at each experimental trial, the cutting process was done in three times under the same cutting condition, NC, p, and Q. After that, the average values were taken.

4) Standard deviations must be included in the results presented and statistical analysis. That a similar analysis is in another article is, of course, mentionable. However, the previous article described a different study, so the results were also different. If this is not the case, is this self-plagiarism?

Answer:

Thank you very much. The statistical analysis was provided and the discussion was expanded in the revised manuscript.

Reviewer 3 Report

The authors have not given a concrete response to many of my comments. For example:

1 - In the introduction the type of lubricant used in the cited works has been omitted:

a) Line 68 - Ali et al. [9] made the study on the tribological characteristics of Al2O3 and TiO2 nanoparticles. Did this work study the influence of these particles in any lubricant? Explain what type of lubricant was used.

b) Line 71 - Yıldırım et al. [10] studied ….. What was the lubricant in this work?

c) Line 74 - Hegab et al. [11] studied MQL performance… What was the lubricant in this work?

d) Line82 - Uysal et al. [14] investigated the tool wear and…. What was the lubricant in this work?

e) And further throughout the Introdction.

2 - It is necessary to present the influence of the factors studied (nanoparticle concentration, air pressure, and air flow rate) on the responses (surface roughness, cutting forces, and coefficient kf).

3 – Las figuras 2, 5, 6, 7 no son informativas.Estas graficas no muestran ninguna relacion de los parametros estudiados (surface roughness, cutting forces, and coefficient kf). Por ejemplo:

a) Figure 2. Influences of cooling and lubricating methods on surface roughness Ra. Where in Figure 2 are the cooling and lubricating methods indicated? Looking at this figure, these methods cannot be found.

b) Figure 5. Effects of investigated variables on the cutting force Fz. What variables are you talking about? Looking at this figure no variable can be found.

c) Figures 6 and 7. The same case as point b).

4 - The method used to determine the temperature in the chip remains unclear. I have seen that most reviewers have asked the same question regarding this method, but none have gotten a convincing answer.

5 – It is a pity, that researchers have not realized that the KEYENCE VHX-7000 is a Digital Microscope and not a Scanning Electron Microscope (SEM). And when I ask in my first report “how can a SEM image indicate what type of color each surface has?” The authors could have corrected the error in figure 3 “SEM images of the machined surface under”. But instead of correcting the error, the authors answered “The SEM images taken on KEYENCE VHX-7000 Digital Microscope can reflect the true color of the machined surface. The authors also changed the images with higher quality so that the color of the machined surface can be easily observed. The deeper discussion was provided and expanded in red color.”

Instead of these images, the authors could use maps of the distribution of elements on the machined surface, thus being able to “see” how the MoS2 and Al2O3 phases are distributed.

In addition to the previous comments, it is necessary to explain:

6 - It is not clear, what is the relation of the sentence on line 63 “In addition, the higher nanoparticle concentration and smaller…[1]” with reference [6]?

7 - Line 66 – Which were the “the base fluids”? What was the lubricant used in this work?

8 – The conclusions must be rewritten based on a new manuscript that must be submitted, so that this material can be published.

Author Response

RESPONSES TO THE REVIEWER 3

We are very grateful for the reviews provided by the editors and each of the external reviewers of this manuscript. Please see below, our detailed response to comments.

The authors have not given a concrete response to many of my comments. For example:

1 - In the introduction the type of lubricant used in the cited works has been omitted:

  1. a) Line 68 - Ali et al. [9] made the study on the tribological characteristics of Al2O3 and TiO2 nanoparticles. Did this work study the influence of these particles in any lubricant? Explain what type of lubricant was used.
  2. b) Line 71 - Yıldırım et al. [10] studied ….. What was the lubricant in this work?
  3. c) Line 74 - Hegab et al. [11] studied MQL performance… What was the lubricant in this work?
  4. d) Line82 - Uysal et al. [14] investigated the tool wear and…. What was the lubricant in this work?
  5. e) And further throughout the Introdction.

Answer

Thank you very much. The lubricant in these works was added and marked in red color.

2 - It is necessary to present the influence of the factors studied (nanoparticle concentration, air pressure, and air flow rate) on the responses (surface roughness, cutting forces, and coefficient kf).

Answer

Thank you very much. The author revised the manuscript to provide the new content and analysis to present the influence of the factors more clearly to convince the readers.

3 – Las figuras 2, 5, 6, 7 no son informativas.Estas graficas no muestran ninguna relacion de los parametros estudiados (surface roughness, cutting forces, and coefficient kf). Por ejemplo:

  1. a) Figure 2. Influences of cooling and lubricating methods on surface roughness Ra. Where in Figure 2 are the cooling and lubricating methods indicated? Looking at this figure, these methods cannot be found.
  2. b) Figure 5. Effects of investigated variables on the cutting force Fz. What variables are you talking about? Looking at this figure no variable can be found.
  3. c) Figures 6 and 7. The same case as point b).

Answer

Thank you very much. The author added more information and the purpose to make and discuss the Figures 2,5, 6. The Figure 7 was removed. Furthermore, the statistical analysis was provided to have the clearer discussion and investigation.

4 - The method used to determine the temperature in the chip remains unclear. I have seen that most reviewers have asked the same question regarding this method, but none have gotten a convincing answer.

Answer

Thank you very much for your valuable comment. The analysis of cutting heat is very important in evaluating the effectiveness of cooling lubrication methods. However, it is difficult to measure heat directly from the cutting process, so here the authors made the indirect assessment through the color of the generated chip. The chip color assessment is based on the color temperature charts. Please allow us to use citations for this analysis.

5 – It is a pity, that researchers have not realized that the KEYENCE VHX-7000 is a Digital Microscope and not a Scanning Electron Microscope (SEM). And when I ask in my first report “how can a SEM image indicate what type of color each surface has?” The authors could have corrected the error in figure 3 “SEM images of the machined surface under”. But instead of correcting the error, the authors answered “The SEM images taken on KEYENCE VHX-7000 Digital Microscope can reflect the true color of the machined surface. The authors also changed the images with higher quality so that the color of the machined surface can be easily observed. The deeper discussion was provided and expanded in red color.”

Instead of these images, the authors could use maps of the distribution of elements on the machined surface, thus being able to “see” how the MoS2 and Al2O3 phases are distributed.

Answer

Thank you very much. The images provided in the manuscipt are microstructure images. The correction was made.

The authors are planning to conduct EDX analysis to demonstrate the distribution of MoS2 and Al2O3 nanoparticles. However, due to a problem with the Scanning Electron Microscope at our university, the photo must be taken at another university. Therefore, we have not done these analyzes yet, so please allow us to cite.

In addition to the previous comments, it is necessary to explain:

6 - It is not clear, what is the relation of the sentence on line 63 “In addition, the higher nanoparticle concentration and smaller…[1]” with reference [6]?

Answer

Thank you very much. The correction of the citation was made.

7 - Line 66 – Which were the “the base fluids”? What was the lubricant used in this work?

Answer

Thank you very much. The base fluids used in this work were added in the revised manuscript.

8 – The conclusions must be rewritten based on a new manuscript that must be submitted, so that this material can be published.

Answer

Thank you very much. The conclusions were rewritten.

Round 3

Reviewer 2 Report

Dear Authors, thank you for your answers and corrections. They are satisfactory.

Author Response

RESPONSES TO THE REVIEWER 2

We are very grateful for the reviews provided by the editors and each of the external reviewers of this manuscript. Please see below, our detailed response to comments.

Dear Authors, thank you for your answers and corrections. They are satisfactory.

Answer:

Thank you very much for your valuable comments and suggestion. Thank you again.

Reviewer 3 Report

I consider that the authors of the work “Machinability Assessment of Hybrid Nano Cutting Oil for Minimum Quantity Lubrication (MQL) in Hard Turning of 90CrSi Steel” have made an effort to correct the errors present in the manuscript. Still, it is possible to find errors (among them serious ones) that need to be corrected. For example.

1 - Figures 2 and 7 do not indicate the concentration of the solid phases for the HF variant

2 – Correct the word SEM between lines 260-266

3 – In figures 4a,b,c it is necessary to add the profilogramm of each one of the surfaces to have a correct appreciation of its quality. A 2D image cannot show how good the roughness obtained is.

4 – Throughout the text there is no description of the methodology to determine the temperature in the chips from the heat of their surfaces. Add to the manuscript a detailed explanation of the used methodology with links to other works where this method has been applied.

5 – Lines 303-308. The work DOI 10.1209/0295-5075/113/36002 (https://iopscience.iop.org/article/10.1209/0295-5075/113/36002/meta) indicates that the thermal conductivity of the MoS2 monolayer is 131 Wm -1 K-1 at 300K, while the work https://doi.org/10.1016/j.matchemphys.2020.123809 (https://www.sciencedirect.com/science/article/pii/S0254058420311688) indicates that the thermal conductivity of nanoalumina is between 0.2-0.3 Wm-1 K-1 This means that the thermal conductivity of nanoalumina is much lower than that of MoS2. From this, it can be seen that your explanation for the presence of the lower temperature in the case of Al2O3 NF is not correct. Give a better explanation of the temperature effect observed in the chips.

6 – Line 312. Use the same format throughout the text 8/2 or 8:2.

7 – Throughout the text and especially in section 3.2.3, correct the abbreviations Fy and Fz.

Explain the names of the abbreviations Fc and Fp. In the manuscript, the abbreviation Fc is called force component and cutting force. Correct it.

Author Response

RESPONSES TO THE REVIEWER 3

We are very grateful for the reviews provided by the editors and each of the external reviewers of this manuscript. Please see below, our detailed response to comments.

I consider that the authors of the work “Machinability Assessment of Hybrid Nano Cutting Oil for Minimum Quantity Lubrication (MQL) in Hard Turning of 90CrSi Steel” have made an effort to correct the errors present in the manuscript. Still, it is possible to find errors (among them serious ones) that need to be corrected. For example.

1 - Figures 2 and 7 do not indicate the concentration of the solid phases for the HF variant

Answer

Thank you very much.

2 – Correct the word SEM between lines 260-266

Answer

Thank you very much. The word is corrected in the revised manuscript.

3 – In figures 4a,b,c it is necessary to add the profilogramm of each one of the surfaces to have a correct appreciation of its quality. A 2D image cannot show how good the roughness obtained is.

Answer

Thank you very much. The analysis images of the corresponding surface profiles for Figures 4a,b,c have been added to support the statements made. Besides, the discussion has been expanded in the revised manuscript.

4 – Throughout the text there is no description of the methodology to determine the temperature in the chips from the heat of their surfaces. Add to the manuscript a detailed explanation of the used methodology with links to other works where this method has been applied.

Answer

Thank you very much. The description of the methodology to indirectly determine the temperature in the chips was added and cited in the revised manuscript.

5 – Lines 303-308. The work DOI 10.1209/0295-5075/113/36002 (https://iopscience.iop.org/article/10.1209/0295-5075/113/36002/meta) indicates that the thermal conductivity of the MoS2 monolayer is 131 Wm -1 K-1 at 300K, while the work https://doi.org/10.1016/j.matchemphys.2020.123809 (https://www.sciencedirect.com/science/article/pii/S0254058420311688) indicates that the thermal conductivity of nanoalumina is between 0.2-0.3 Wm-1 K-1 This means that the thermal conductivity of nanoalumina is much lower than that of MoS2. From this, it can be seen that your explanation for the presence of the lower temperature in the case of Al2O3 NF is not correct. Give a better explanation of the temperature effect observed in the chips.

Answer

Thank you very much. Reviewer's comment is correct. The explanation has been rewritten and the basis for this is cited in the revised manuscript.

6 – Line 312. Use the same format throughout the text 8/2 or 8:2.

Answer

Thank you very much. The text is corrected to the same format in the revised manuscript.

7 – Throughout the text and especially in section 3.2.3, correct the abbreviations Fy and Fz.

Answer

Thank you very much. The abbreviations Fy and Fz  were corrected in the revised manuscript.

8- Explain the names of the abbreviations Fc and Fp. In the manuscript, the abbreviation Fc is called force component and cutting force. Correct it.

Answer

Thank you very much. The abbreviations Fc and Fp were explained and corrected with red color in the revised manuscript.

Round 4

Reviewer 3 Report

I have the impression that the experiments and temperature measurements were carried out incorrectly, and it is not possible to repeat this study by another scientific group since there is no necessary and adequate information. For this reason, I considered that this article should be rejected so that the authors rewrite it correctly and once it is completely ready, they can resubmit it for publication in this journal.

1 – In the tables of figures 2, 6 and 7 the concentration of the solid phases for the HF MQL variant is not yet indicated. Please, add them.

2 – the profilogramms in figures 4a,b,c are not legible or understandable. Values on the color scale cannot be read. The same happens with the roughness values in the table and the values in the profile graphs. The profile of the surface is indistinguishable from the black background. Draw the profile with a thicker line and white color. Increase the size of the graphics to make them understandable.

3 – In the manuscript you can still find the abbreviations Fy and Fz, Please, find and correct it.

4 – The links related to the temperature assessment method do not give the possibility of knowing and studying the method used. For example, the ref. [27] is to a closed access article. And in the ref. [31] there is no description of the method used and even less of a temperature table that can be used to compare the temperature values. I recommend adding links to publications of other scientific groups that have used this method and to color tables obtained for the 90CrSi Steel material.

5 - In my comment 5 of my previous review I asked to give a better explanation of the temperature effect observed in the chips, but the authors now wrote something that is less convincing since they contradict themselves. For example, the authors wrote “Al2O3 nano cutting fluid exhibits the better effects on reducing cutting forces and surface roughness in this case.” while figure 2 shows that the roughness obtained with Al2O3 nano cutting fluid is higher than the roughness obtained with MoS2 nano cutting oil in 60% of the experiments. Likewise, figure 6 shows that in 73% of the cases the cutting force with the Al2O3 nano cutting fluid is higher than the cutting force recorded with the MoS2 nano cutting oil in 60% of the experiments.

6 - Furthermore, the authors write “As a result, the heat on the underside of the chip is lower when compared to MoS2 nano cutting oil [31].” But in this reference there is no information that corroborates this statement made.

Author Response

RESPONSES TO THE REVIEWER 3

We are very grateful for the reviews provided by the editors and each of the external reviewers of this manuscript. Please see below, our detailed response to comments.

I have the impression that the experiments and temperature measurements were carried out incorrectly, and it is not possible to repeat this study by another scientific group since there is no necessary and adequate information. For this reason, I considered that this article should be rejected so that the authors rewrite it correctly and once it is completely ready, they can resubmit it for publication in this journal.

1 – In the tables of figures 2, 6 and 7 the concentration of the solid phases for the HF MQL variant is not yet indicated. Please, add them.

Answer:

Thank you very much. The concentration of the solid phases for the HF MQL variant was added to the tables of figures 2, 6 and 7

2 – the profilogramms in figures 4a,b,c are not legible or understandable. Values on the color scale cannot be read. The same happens with the roughness values in the table and the values in the profile graphs. The profile of the surface is indistinguishable from the black background. Draw the profile with a thicker line and white color. Increase the size of the graphics to make them understandable.

Answer:

Thank you very much. The image quality of figures 4a,b,c was improved and put in Figure 5 in revised manuscript in order to observe clearly. The profile lines are the original images taken from VHX 7000, so the line thickness and background color cannot be adjusted. However, on the newly replaced pictures, the quality has improved and looks very clear.

3 – In the manuscript you can still find the abbreviations Fy and Fz, Please, find and correct it.

Answer:

Thank you very much. The abbreviations Fy and Fz were corrected.

4 – The links related to the temperature assessment method do not give the possibility of knowing and studying the method used. For example, the ref. [27] is to a closed access article. And in the ref. [31] there is no description of the method used and even less of a temperature table that can be used to compare the temperature values. I recommend adding links to publications of other scientific groups that have used this method and to color tables obtained for the 90CrSi Steel material.

Answer:

Thank you very much. After revising the reviewer’s comment, the content of indirect evaluation of cutting heat through chip color was removed from the content of the article.

5 - In my comment 5 of my previous review I asked to give a better explanation of the temperature effect observed in the chips, but the authors now wrote something that is less convincing since they contradict themselves. For example, the authors wrote “Al2O3 nano cutting fluid exhibits the better effects on reducing cutting forces and surface roughness in this case.” while figure 2 shows that the roughness obtained with Al2O3 nano cutting fluid is higher than the roughness obtained with MoS2 nano cutting oil in 60% of the experiments. Likewise, figure 6 shows that in 73% of the cases the cutting force with the Al2O3 nano cutting fluid is higher than the cutting force recorded with the MoS2 nano cutting oil in 60% of the experiments.

Answer:

Thank you very much. After revising the reviewer’s comment, the content of indirect evaluation of cutting heat through chip color was removed from the content of the article.

6 - Furthermore, the authors write “As a result, the heat on the underside of the chip is lower when compared to MoS2 nano cutting oil [31].” But in this reference there is no information that corroborates this statement made.

Answer:

Thank you very much. After revising the reviewer’s comment, the content of indirect evaluation of cutting heat through chip color was removed from the content of the article.
